# Mechanical force regulates ligand binding and function of PD-1

Kaitao Li [1,2,7], Paul Cardenas-Lizana[1,2,8], Jintian Lyu[1,2,9], Anna V. Kellner[1,10], Menglan Li [1,2], Peiwen Cong [1,2], Valencia E. Watson[1,2], Zhou Yuan[1,2,3,11], Eunseon Ahn[4,5,12], Larissa Doudy[1,2], Zhenhai Li [1,3,13], Khalid Salaita [1,6], Rafi Ahmed [4,5] & Cheng Zhu [1,2,3] ✉

Despite the success of PD-1 blockade in cancer therapy, how PD-1 initiates signaling remains unclear. Soluble PD-L1 is found in patient sera and can bind PD-1 but fails to suppress T cell function. Here, we show that PD-1 function is reduced when mechanical support on ligand is removed. Mechanistically, cells exert forces to PD-1 and prolong bond lifetime at forces <7 pN (catch bond) while accelerate dissociation at forces >8pN (slip bond). Molecular dynamics of PD-1–PD-L2 complex suggests force may cause relative rotation and translation between the two molecules yielding distinct atomic contacts not observed in the crystal structure. Compared to wild-type, PD-1 mutants targeting the force-induced distinct interactions maintain the same binding affinity but suppressed/eliminated catch bond, lowered rupture force, and reduced inhibitory function. Our results uncover a mechanism for cells to probe the mechanical support of PD-1–PD-Ligand bonds using endogenous forces to regulate PD-1 signaling.

Programmed cell death-1 (PD-1) is an immune checkpoint receptor and a hallmark of T cell exhaustion in chronic viral infection and cancer[1–3]. PD-1 consists of a single IgV domain, a ~ 20 AA stalk, a transmembrane region, and an intracellular tail with two tyrosine-based signaling motifs: an immunoreceptor tyrosine-based inhibitory motif (ITIM) and an immunoreceptor tyrosine-based switch motif (ITSM)[4]. Binding of PD-1 to its ligand (PD-L1 or PD-L2) triggers the phosphorylation of ITIM and ITSM, which recruits and activates SH2-containing tyrosine phosphatase 2 (SHP-2). Activated SHP-2 dephosphorylates a panel of signaling molecules downstream of TCR and CD28, thereby suppressing T cell activation and function[5,6]. Blocking antibodies targeting PD-1–PD-L1 interaction have yielded great success in cancer

immunotherapy[7,8]. However, for a molecule like PD-1 with such a simple structure, the detailed molecular mechanism as how it transduces ligand binding to initiate signaling remains unclear. It becomes even more puzzling that as a prognosis marker of various types of cancer, some soluble PD-L1 splicing variants retain their ability of PD-1 binding but fail to trigger suppression of T cells[9–12]. We also found in this study that soluble PD-Ligands, even in tetrameric forms, are ineffective in inducing PD-1 function whereas cell surface or bead-coated PD-Ligands delivered robust PD-1 triggering.

It is worth noting that one of the critical components missing in the soluble form compared with the surface-anchored form of ligands is the ability to provide physical support to the engaged PD-1–PD-

[1]Wallace H. Coulter Department of Biomedical Engineering, Atlanta, GA, USA. [2]Parker H. Petit Institute for Bioengineering and Biosciences, Atlanta, GA, USA. [3]George W. Woodruff School of Mechanical Engineering, Georgia Institute of Technology, Atlanta, GA 30332, USA. [4]Emory Vaccine Center, Atlanta, GA, USA. [5]Department of Microbiology and Immunology, Emory University School of Medicine, Atlanta, GA, USA. [6]Department of Chemistry, Emory University, Atlanta, GA 30322, USA. [7]Present address: Shennon Biotechnologies, San Francisco, CA, USA. [8]Present address: Department of Bioengineering and Chemical Engineering, University of Engineering and Technology—UTEC, Lima, Peru. [9]Present address: L.E.K. consulting, Boston, MA, USA. [10]Present address: Elephas, Madison, WI, USA. [11]Present address: Alfred E. Mann Department of Biomedical Engineering, University of Southern California, Los Angeles, CA, USA. [12]Present address: Merck, South San Francisco, CA, USA. [13]Present address: School of Mechanics and Engineering Science, Shanghai University, Shanghai, China. ✉e-mail: cheng.zhu@bme.gatech.edu

Ligand bond for it to bear mechanical load. Using DNA-based molecular tension probes (MTP) with a locking strand to accumulate the force signals we have shown that activated T cells actively apply forces on PD-1 engaged with PD-L2 or anti-PD-1 antibody[13]. In this study we also observed significant tension using MTP of 4.7 pN threshold force presenting PD-L1 or PD-L2. The formation and movement of PD-1 microclusters[5,14] as well as the ability of PD-1–PD-L2 interaction to drive "synapse"-like interface between CHO cells[15] also suggest PD-1–PD-Ligand bonds are constantly under mechanical load. Emerging evidence shows that mechanical force plays a critical role in immune cell functions by modulating the binding and signaling of various immune receptors[16–19]. For example, force can enhance antigen recognition by TCR and BCR and potentiate target killing by effector T cells[20–25]. At the molecular level, TCR antigen recognition is enhanced by the dynamic responses of TCR–pMHC interaction to force application – potent ligands form catch bonds with more durable TCR engagement and signaling, whereas weak ligands form slip bonds that readily rupture under force[20,22,26–29]. Together, these findings raise the intriguing question of the role of force on PD-1 ligand binding and function.

Here, we investigated the effect of force by comparing PD-1 inhibitions on TCR signaling and function through engagement with surface-bound *vs* soluble PD-Ligands, finding mechanical support enhances PD-1 inhibitory function. We employed MTP to report endogenous force and biomembrane force probe (BFP) to measure in situ kinetics, finding that T cells pull on PD-1 with forces between 4.7 and 12 pN and applied force elicits catch-slip bonds for both PD-1–PD-L1 and PD-1–PD-L2 interactions, where forces below 7 pN prolongs bond lifetime (catch) and forces above 8 pN accelerates dissociation (slip). Corroborating the force-enhanced bonding, in silico simulations using steered molecular dynamics (SMD) suggests that pulling on the two C-termini of the PD-1–PD-L2 complex induces large relative rotation and translation between the two molecules, which is accompanied by formation of distinct atomic contacts not observed in the crystal structure in absence of applied force. Mutating residuals on PD-1 involved in the force-induced distinct non-covalent interactions decreased the mechanical stability of the PD-1–PD-L2 bond manifesting a lower force required for bond rupture, shorter bond lifetime under sustained force, and reduced the number of bonds bearing above threshold endogenous forces, despite the lack of effect of the mutations on the in situ PD-1–PD-L2 binding affinity in the absence of force. Most importantly, these PD-1 mutants demonstrate impaired ability to suppress TCR-CD3 induced NFκB activation. Overall, our results suggest force critically regulates the mechanical stability of PD-1–PD-Ligand bond, which is essential for efficient PD-1 triggering.

## Results

### The inhibitory function of PD-1 is enhanced by ligand bearing mechanical support

Recent studies have found soluble forms of alternatively spliced PD-1 ligands as prognosis biomarkers in various cancers[10,30]. Despite their ability to bind to PD-1, results were discrepant regarding whether soluble PD-1 ligands can induce immunosuppression[9,11,12,31]. To resolve this, we compared PD-1 function triggered by PD-1 ligands expressed on cell membrane or immobilized on beads versus in soluble tetrameric forms. NFκB::eGFP reporter Jurkat cells[32] were transduced to overexpress a chimeric PD-1 molecule consisting of mouse PD-1 ectodomain (Met1 to Met169) and human PD-1 transmembrane and intracellular domain (Val171 to Leu288) (Fig. S1A). GFP expression can be induced by stimulating the cells with anti-CD3 (clone OKT3) full antibody or T-cell stimulator cells (TSC) that expresses a membrane-anchored single-chain fragment variable (scFv) of OKT3[33]. Plain and PD-1 reporter Jurkat cells were cocultured with control or PD-Ligand expressing TSCs (Fig. S1B, C) and analyzed for their GFP expression (Fig. S1D). As expected, cell membrane PD-L1 or PD-L2 triggered potent inhibition of OKT3-induced GFP expression as quantified by the

frequency of GFP+ PD-1 reporter cells and their geometric mean fluorescence intensity (gMFI) (Fig. S1E–G). To avoid altering anti-CD3 presentation when comparing immobilized *vs* soluble PD-Ligands, plain or PD-1 reporter cells were stimulated with soluble OKT3 together with soluble (Fig. 1A–C) or bead-coated (Fig. 1D–F) PD-Ligands and analyzed for their GFP expression. Comparing to control group of anti-CD3 plus soluble streptavidin (SA), the experimental groups of anti-CD3 plus PD-L1 or PD-L2 tetramer failed to induce suppression of GFP induction in PD-1 reporter cells just like in plain control cells, despite the high concentration (20 μg/ml based on monomer) used (Fig. 1A–C). However, when coated on beads, both PD-L1 and PD-L2 triggered potent inhibition of anti-CD3 induced GFP expression in PD-1 reporter cells but not in plain control cells (Fig. 1D–F). Together, these results indicated that successful triggering of PD-1 requires cell membrane or surface anchored ligands, regardless of their ability to cluster by crosslinking. Interestingly, such a requirement is not limited to natural ligands of PD-1. When PD-Ligands in Fig. 1A–F were replaced with anti-PD-1 antibodies, all three clones (29 F.1A12, J43, and RMP1-30) show no effect of PD-1 triggering in soluble tetramer form but strong effect on PD-1 triggering when immobilized on beads (Fig. S2), suggesting surface anchoring is essential to the PD-1 agonist effect.

Given that force has been shown to have critical effect in the triggering of other membrane receptors[16–19], the above data prompt us to hypothesize that one critical component for PD-1 triggering is mechanical force on PD-1, which can be generated endogenously by T cells and supported by engaged ligands or antibodies if they are anchored on a solid surface to provide counter-balance force[13]. To further test this hypothesis, we altered the bead-based ligand presentation system to modulate its ability to support mechanical forces on PD-L1/L2. By introducing a [PEG]24 spacer between the bead surface and SA, a - 10 nm extra length was added to the ligand, which is comparable to the dimension of a PD-1–PD-Ligand spanned across the intercellular junction[15]. This should prevent at least part of the PD-1–PD-Ligand bonds to be fully stretched due to the ligand elongation, but should not affect PD-1–PD-Ligand binding (Fig. 1G, H). When both bead-coated PD-Ligands were tested against PD-1 reporter Jurkat cells under anti-CD3 stimulation, PD-Ligands with a [PEG]24 spacer induced less inhibition comparing to those without the spacer (Fig. 1I–J), supporting our hypothesis because PD-1 triggering was reduced by dampening the mechanical support on PD-Ligands.

To further confirm the hypothesis using primary T cells, we activated CD8+ OT1 T cells in vitro to induce PD-1 expression and then tested its inhibition of SIINFEKL:H2-Kb triggered cell spreading and calcium signaling upon engagement with either soluble or bead-coated PD-Ligand (Fig. 2). Both readouts are subject to PD-1's inhibition according to our previous study using P14 CD8 T cells triggered by LCMV gp33:H2-Db pMHC[34]. To ensure that we only modulate the mechanical support on PD-Ligand, we presented pMHC on glass surface to stimulate OT1 T cells while engaging PD-1 by incubating cells with either soluble PD-Ligand tetramer or PD-Ligand-coated beads (Fig. 2A, D). OT1 T cells bound with PD-Ligand tetramer or beads were monitor for their spreading on pMHC-coated glass cover slip using Reflection Interference Contrast Microscopy (RICM) (Fig. 2B) or calcium flux using X-rhod-1calcium indicator (Fig. 2E). Consistent with the results using Jurkat cells, significant inhibition of pMHC-triggered cell spreading (Fig. 2B) or calcium flux (Fig. 2F) were only induced by PD-Ligand immobilized on beads but not tetramers in solution. In another setup where OT1 T cells were stimulated by repeated touches with a pMHC-coated human Red Blood Cell (RBC)[35] (Fig. 2G and Movie S1), contacting PD-1 with a PD-L1-coated bead but not BSA-coated bead and soluble PD-L1 tetramer significantly suppressed the pMHC-triggered calcium flux, and contacting PD-1 with a PD-L2-coated bead suppressed the pMHC-triggered calcium flux more significantly than BSA-coated bead and soluble PD-L2 tetramer (Fig. 2H). Together, these data further

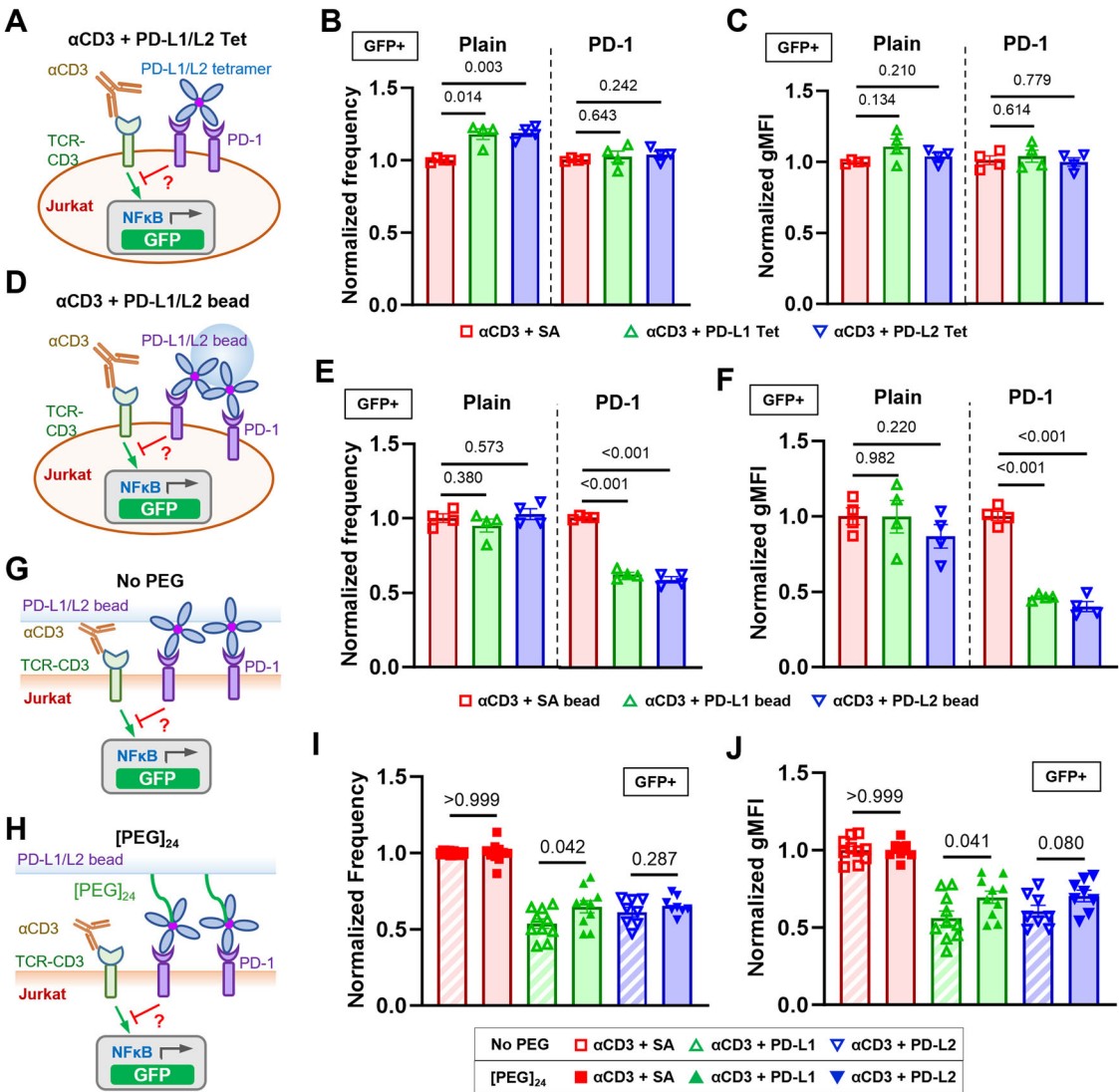

Fig. 1 | The inhibitory function of PD-1 on Jurkat cells is enhanced by mechanical support of PD-Ligands. A Schematics of stimulating NFκB::eGFP reporter Jurkat cells with soluble anti-CD3 and soluble PD-ligand tetramers. B, C Quantification of GFP expression for condition in (A). $n = 4$ for all conditions pooled from two independent experiments. D Schematics of stimulating NFκB::eGFP reporter Jurkat cells with soluble anti-CD3 and PD-ligand-coated beads. E, F Quantification of GFP expression for condition in (B). $n = 4$ for all conditions pooled from two independent experiments. Schematics of stimulating NFκB::eGFP reporter Jurkat cells with soluble anti-CD3 and PD-Ligands coated beads without (G) or with (H) [PEG]$_{24}$ spacer arm. I, J Quantification of GFP expression for conditions in (G, H). $n = 10$, 10, and 8 for SA, PD-L1, and PD-L2, respectively, pooled from 5 independent experiments. Normalized frequency (B, E, I) and normalized geometric mean fluorescence intensity (gMFI) (C, F, J) were calculated as (sample−averaged background)/(anti-CD3 control−averaged background) and presented as mean ± SEM. Numbers on graphs represent p values calculated from two-tailed student $t$ test. Source data are provided in Source Data file.

emphasize the importance of mechanical support on PD-Ligand for efficient PD-1 function. This phenomenon was also observed when OT1 T cells were tested in B16F10 melanoma cell line conditioned media (Fig. S3), suggesting it's not affected by the soluble factors from tumor cell culture.

**Molecular tension probes reveal active cellular forces applied to PD-1−PD-Ligand bonds**

We next used MTP-tagged PD-L1 or PD-L2 to directly measure forces applied to single PD-1−PD-ligand bonds by CHO cells overexpressing PD-1 (Fig. S4A). The MTP consists of three DNA strands linking a Cy3B fluorophore to the PD-Ligand and a BHQ2 quencher to the glass surface, which are brought together by a hairpin structure to control the force threshold by varying length and GC content[36,37]. Forces on PD-1−PD-Ligand that are above the threshold unfold the hairpin, thereby separating the Cy3B from the BHQ2 to enable fluorescence (Fig. S4A). RICM images show that CHO cells spread similarly on 4.7

and 12 pN MTPs conjugated with both PD-Ligands, but epifluorescence imaging reveal significantly higher Cy3B signals from the 4.7 than the 12 pN MTPs (Fig. S4 and Movie S2), yet both cell spreading and tension were nearly eliminated by an anti-PD-1 blocking antibody (Fig. S4B–D). These results indicate that CHO cell spreading and pulling are mediated by specific PD-1−PD-Ligand interaction. While the spreading areas are similar for PD-L1 vs PD-L2 or MTPs with 4.7 pN vs 12 pN threshold forces (Fig. S4B, C), the tension signals were stronger for PD-L2 than PD-L1 with 4.7 pN MTP but the difference vanished as the tension signals for both PD-Ligands decreased significantly when the MTP's force threshold was increased to 12 pN (Fig. S4B, D).

To confirm the observed forces were not characteristics of the adherent phenotype of CHO cells, the same experiment was repeated with OT1 T cells in vitro activated to induce PD-1 expression. Due to the low expression of PD-1, the DNA hairpin was linked to the surface through a gold nanoparticle to further quench the Cy3B to increase

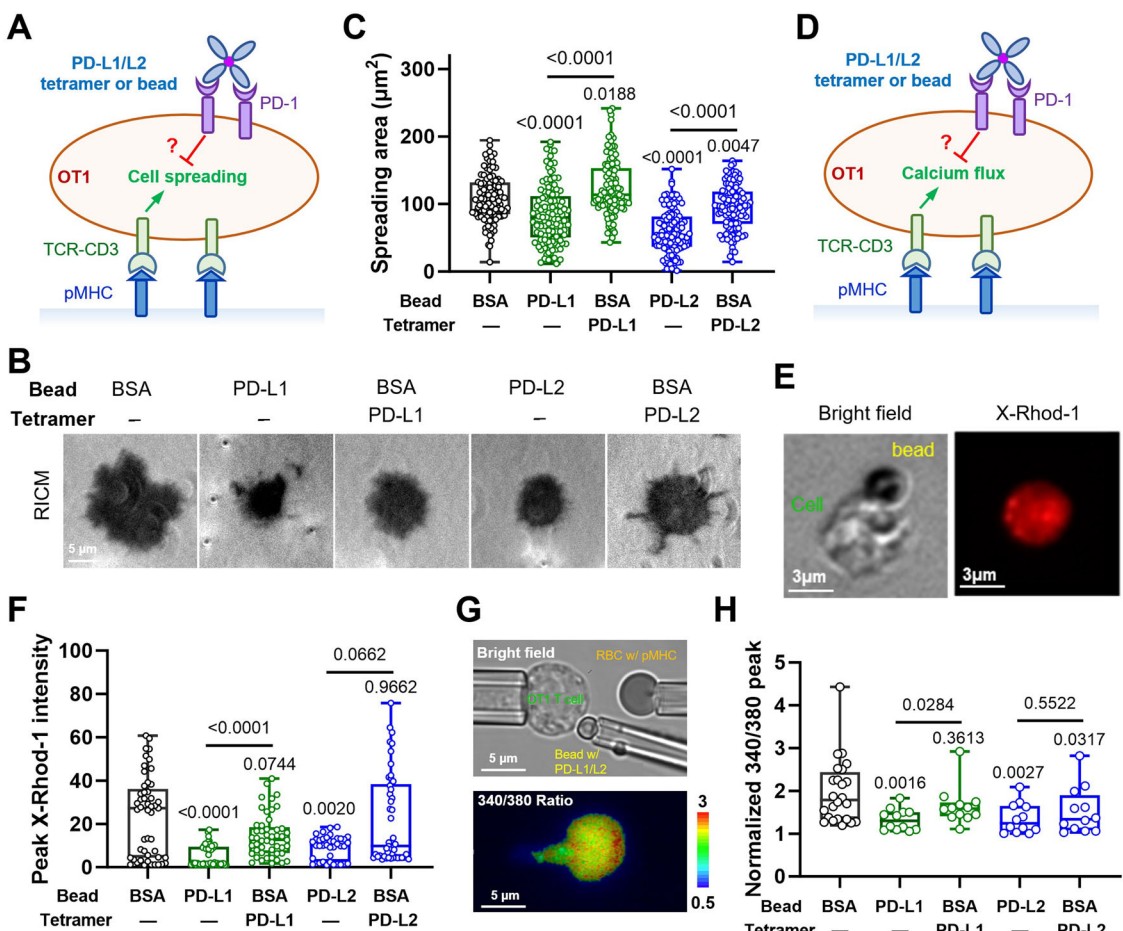

**Fig. 2 | The inhibitory function of PD-1 on activated primary T cells is enhanced by mechanical support of PD-Ligands. A** Schematics of measuring the suppression of pMHC (SIINFEKL:H2-K$^b$)-mediated OT1 T cell spreading by soluble PD-Ligand tetramer or PD-Ligand-coated beads. **B** Representative reflection interference contrast microscopy (RICM) images of an activated OT1 T cell spreading on glass surface functionalized with SIINFEKL:H2-K$^b$ under indicated conditions. **C** Quantification of cell spreading area for conditions in (**B**). $n = 114, 111, 115, 116$, and 114 cells pooled from 2 independent experiments. **D** Schematics of measuring the suppression of pMHC (SIINFEKL:H2-K$^b$)-mediated OT1 T cell calcium flux by soluble PD-Ligand tetramer or PD-Ligand-coated beads. **E** Representative bright-field and X-Rhod-1 images of SIINFEKL:H2-K$^b$ surface stimulation of an OT1 T cell bound to PD-L1-coated beads. **F** Quantification of peak X-Rhod-1 fluorescence under

indicated conditions for experiments illustrated in (**D**, **E**). $n = 49, 49, 49, 38$, and 49 cells. **G** Representative bright-field (*upper*) and Fura-2 340/380 radiometric pseudocolor (*lower*) images illustrating measurement of the suppression of pMHC (SIINFEKL:H2-K$^b$)-mediated OT1 T cell calcium flux by soluble PD-Ligand tetramer or PD-Ligand-coated beads using a fluorescence micropipette adhesion frequency (fMAF) setup using three micropipettes. See also Movie S1. **H** Quantification of peak Fura-2 340/380 ratios under indicated conditions for experiments illustrated in (**G**). $n = 24, 12, 12, 12$, and 12 T cell-RBC pairs pooled from 2 independent experiments. Data were presented in box (median with 25%/75% boundaries) and whisker (min and max) plots. Numbers on graphs represent p values calculated from two-tailed Mann–Whitney U test of indicated two groups or the experiment group (green or blue) with BSA control (black). Source data are provided in Source Data file.

---

the signal-to-noise[36], and a complementary strand of DNA to the unfolded hairpin was added the solution to lock it in the opened configuration, which allows the tension signals to be accumulated over time (Fig. 3A)[13]. Consistent with the results obtained using CHO cells transfected PD-1, primary T cells also pulled on endogenous PD-1. Also, stronger tension signals were observed for PD-L2 than PD-L1 as well as for MTPs with 4.7pN than 12pN threshold force (Fig. 3B–D).

## PD-1 forms catch bond with PD-L1 and PD-L2
To test the hypotheses that PD-1–PD-Ligand bonds can sustain forces between 4.7 pN and 12 pN and that PD-1–PD-L2 bonds are more mechanically stable than PD-1–PD-L1 bonds, we performed dynamic force spectroscopic analysis of the rupture forces (force-ramp spectroscopy) and bond lifetimes (force-clamp spectroscopy) of PD-1 expressed on CHO cells interacting with PD-L1 or PD-L2 using a BFP (Fig. 4A)[38]. A CHO cell was repetitively brought into contact with a BFP bead coated with PD-Ligand and then separated until the bond ruptured (Fig. 4B) or held at a preset force until spontaneous

dissociation (Fig. 4C). The magnitude of force is calibrated from the spring constant (-0.3 pN/nm) of the BFP and the displacement of the bead, which was tracked with millisecond temporal resolution and nanometer spatial resolution, resulting in pico-newton force resolution[39]. Hundreds of bonding events were analyzed for PD-1 interacting with each PD-Ligand and were pooled to analyze the rupture force distribution at a nominal loading rate of 1000 pN/s (Fig. 4D, E) or bond lifetime distribution around 7 pN forces (Fig. 4F). The cumulative frequency of ruptured events *vs* force at which bond ruptures follows a sigmodal shape with the PD-1–PD-L2 curve right-shifted towards higher forces relative to the PD-1–PD-L1 curve (Fig. 4E). The force at which 50% of the bonds rupture ($F_{1/2}$) was 12.1 pN for PD-1–PD-L2 bonds, significantly higher than the 10.3 pN value for the PD-1–PD-L1 bonds.

Interestingly, the mean ± sem lifetime *vs* force curves of PD-1–PD-L1 and PD-1–PD-L2 interactions both display a catch-slip bond characteristics, where bond lifetime increased with force below 7 pN ("catch") and decreased with force above 8 pN ("slip") (Fig. 4G), a phenomenon observed for many other molecular interactions[19,40].

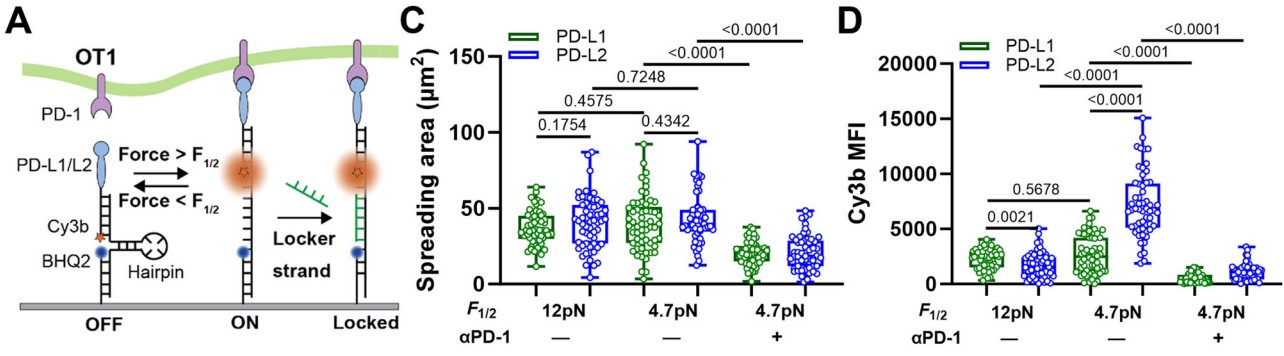

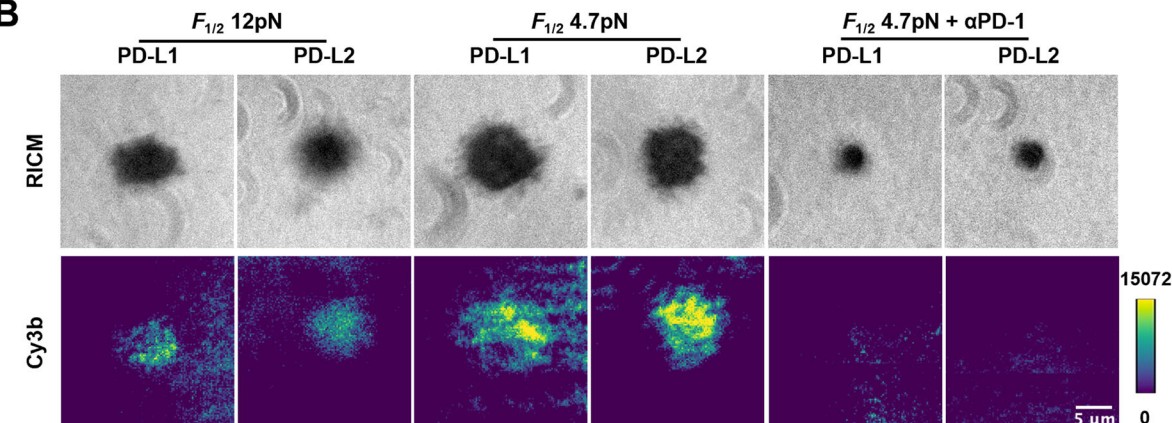

**Fig. 3 | DNA-based MTPs reveal OT1 T cells applying forces to PD-1–PD-Ligand bonds. A** Schematics of visualizing endogenous forces on PD-1–PD-Ligand bonds using DNA-based molecular tension probes (MTPs). Forces above the force threshold unfold the hairpin to separate Cy3B from the BHQ2, which de-quenches the fluorophore. A complementary single stranded DNA (locker) in solution hybridizes with the unfolded hairpin to lock it in the open configuration, which enables accumulation of the fluorescence signals over time. **B** Representative reflection interference contrast microscopy (RICM) and total internal reflection

fluorescence (TIRF) images of activated OT1 T cells interacting with glass surface functionalized with MTPs of indicated conditions. For PD-1 blockade, cells were pre-incubated with PD-1 blocking antibody clone 29 F.1A12 before imaging. Quantification of cell spreading area (**C**) and Cy3b fluorescence (**D**) for conditions in (**B**). $n = 57, 59, 58, 57, 60,$ and $58$ pooled from 1 in 3 independent experiments. Data were presented in box (median with 25%/75% boundaries) and whisker (min and max) plots. Numbers on graphs represent $p$ values calculated from two-tailed Mann–Whitney U test. Source data are provided in Source Data file.

Plotting the pooled bond lifetime distribution of each force bin reveals a dynamic composition of short (<0.1 s), intermediate (0.1 s to 1 s), and long (>1 s) lifetime species in response to force, suggesting multiple states during force-induced dissociation (Figs. S5, S6). Consistent with the higher rupture forces of the PD-1–PD-L2 bond than the PD-1–PD-L1 bond, the lifetime *vs* force curve for the PD-1–PD-L2 bond was up-shifted toward longer lifetime across different force levels relative to the curve for the PD-1–PD-L1 bond (Fig. 4G). These results support our hypotheses and suggest that PD-1–PD-L2 bond is more mechanically stable than PD-1–PD-L1 bond, which is consistent with the higher tension signal mediated by PD-1–PD-L2 interaction than PD-1–PD-L1 interaction (Figs. 3D, S4D).

## SMD reveals force-induced conformational changes in PD-1–PD-L2 complex and formation of distinct non-covalent contact at the atomic level

The structures of PD-1–PD-Ligand complexes of human and mouse species all show a "side-to-side" interaction of β-sheets from two immunoglobular domains, manifesting an assembly similar to the variable domains of α and β chains of TCR or heavy and light chains of antibody[15,41,42]. In such assembly positions, PD-1 and PD-L2 form complex with a sharp angle between their respective long axes, providing a lever arm for the tensile force applying at the C-termini of the two molecules to exert a moment to unbend this angle (Fig. 5A). To gain structural insights of the effect of force on PD-1–PD-

Ligand complex, we applied free molecular dynamics (FMD) and steered molecular dynamics (SMD) to simulate the dynamic responses of the PD-1–PD-L2 structure (PDB: 3BP5) without force or with force applied to the C-termini of the two molecules, respectively. Atomic coordinates were tracked for a total of 60 ns and analyzed for overall structural changes (Fig. 5A, B) and atomic-level bonding/debonding events (Fig. 5C–H). Snapshots of the complex at different time points show that force may gradually aligned the two molecules along their long axes converting the "side-to-side" interaction to a nearly "head-to-head" position before bond rupture (Fig. 5A and Movie S3). This putative conformational change manifests large changes in both the relative angle (from ~50° to -160°) and displacement (-30 Å measured by RMSD) of the two molecules with distinct transitions among multiple phases (Fig. 5A, B), which may explain the dynamic composition of short, intermediate, and long bond lifetimes under force (Figs. S5, S6).

Detailed analysis of bonding/debonding events at atomic scale reveals potential force-induced changes of different types of interactions underlying the binding interface rearrangement in SMD, which was absent in FMD. During SMD the total number of hydrogen bond (H-bond) was stable in phase I and shifted to an overall down trend in phase II with brief but significant pull backs in phases III and IV (Fig. 5C). Salt bridge interactions were enhanced by force in late phase I through phase III before being suppressed in phase IV (Fig. 5D). In contrast, total number of hydrophobic contacts remains unchanged

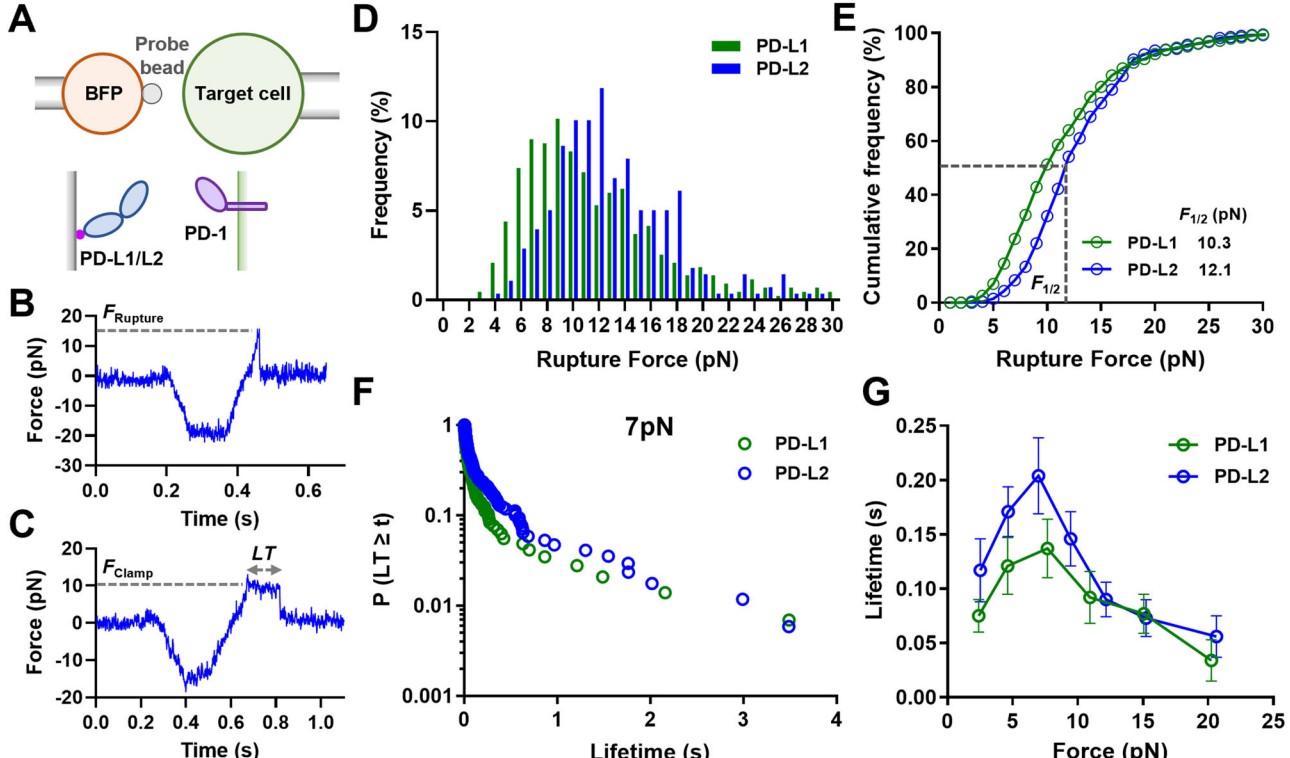

**Fig. 4 | PD-1 forms catch bond with PD-L1 and PD-L2. A** Schematics of force spectroscopic analysis of PD-1–PD-Ligand bonds using biomembrane force probe (BFP). CHO cells expressing PD-1 were analyzed against BFP bead coated with PD-L1 or PD-L2. Bead displacements were tracked with high spatiotemporal resolution and translated into force after multiplying by the spring constant of BFP. Representative raw traces of rupture force (**B**) and bond lifetime (**C**) measurements. A target cell held by piezo-driven micropipette was brought into brief contact with a bead (approach and contact) to allow for bond formation. Upon separation the target cell either kept retracting to rupture the bond (**B**) or stopped and held at a predefined force level until bond dissociated spontaneously (**C**). Force histograms (**D**) and cumulative frequencies (**E**) of rupture events of 433 PD-1–PD-L1 and 278 PD-1–PD-L2 bonds. $F_{1/2}$ is defined as the force level at which 50% of the bonds are ruptured. $p < 0.0001$ comparing $F_{1/2}$ of PD-1–PD-L1 and PD-1–PD-L2 using two-tailed Mann–Whitney test. Survival frequencies at the 7 pN force bin (**F**) and mean ± sem bond lifetime vs force plots (**G**) of PD-1–PD-L1 ($n = 55, 129, 144, 120, 33,$ and 16 lifetime events) and PD-1–PD-L2 ($n = 29, 165, 170, 173, 105, 82,$ and 53 lifetime events) bonds. $p < 0.0001$ comparing lifetime vs force distributions of PD-1–PD-L1 and PD-1–PD-L2 using two-tailed two-dimensional Kolmogorov-Smirnov test. Source data are provided in Source Data file.

until increases by ~50% in phase IV (Fig. 5E). Together, these observations suggest that applied force may induce formation of different types of distinct noncovalent contacts while disrupting the original ones, apparently rendering reinforcement during certain phases in the dissociation dynamics when combining all atomic interactions together. In particular, we noticed that some of the putative force enhanced atomic contacts were not located in the binding pocket of the crystal structure or disrupt force-free PD-1–PD-L2 binding (PD-L2 Ig staining) when mutated[15], such as Leu128, Lys131 and Ala132 located in the FG loop of PD-1 (Fig. 5F–H). These observations prompted us to hypothesize that one of their roles is to stabilize PD-1–PD-L2 bond under force.

**PD-1 mutants targeting force-induced atomic contacts impairs PD-1–PD-L2 mechanical stability**

To test the aforementioned hypothesis, we made single- and double-residue PD-1 mutations K131A, L128A/K131A, and A132K (Fig. 5F–H) and expressed these mutants in CHO cells and NFκB::eGFP reporter Jurkat cells at similar levels (Fig. S7) to test the consequences of eliminating the atomic interactions formed by these residues with PD-L2 under force. When analyzed against RBCs coated with PD-L2, all three PD-1 mutants show similar two-dimensional (2D) effective affinity as WT PD-1 (Fig. 6A), consistent with previous studies showing similar PD-L2 Ig staining[15]. However, dynamic force spectroscopic analysis with BFP showed that all three mutants left-shifted the cumulative frequency curves towards lower forces relative to the WT curve (Fig. 6B). Also, the

two single-mutants down-shifted the bond lifetime vs force curves towards shorter lifetimes relative to the WT curve, and the double-mutant completely converted the catch bond to a slip bond (Fig. 6C). Evaluating the combined effect of ruptured bonds (rupture events) and unruptured bonds (lifetime events) as multiplying bond lifetime by the probability of bond survival at the corresponding force level (1 − cumulative frequency of bond rupture), we estimated the average effective bond lifetime as what would be sampled by PD-1 on cells at given force levels, which modified the bond lifetime vs force curves quantitatively but not qualitatively (Fig. 6D). Together, these results show that the mutations weaken the force-enhanced PD-1–PD-L2 bond stability, which supported our hypothesis and revealed the structural mechanisms underlying the mechanical stability of the PD-1–PD-L2 bond under force.

To directly visualize the effect of these mutations on PD-1–PD-L2 bonds under active cellular forces, we analyzed the cell spreading and tension using PD-L2 coupled MTP of 4.7 pN threshold force. Consistent with impaired bond strength and lifetime, both cell spreading and tension signal was reduced in CHO cells (Fig. 6E–G) or NFκB::eGFP reporter Jurkat cells (Fig. 6H–J) expressing these mutants with the level of impairment ranking as K131A < L128A/K131A < A132K (Fig. 6EJ). Since the 2D affinity of the mutants are similar to that of WT, such reductions in spreading and tension indicate that the reduced stability of bonds between PD-L2 and PD-1 mutants has decreased the numbers of opened MTPs or become less able to support cell spreading.

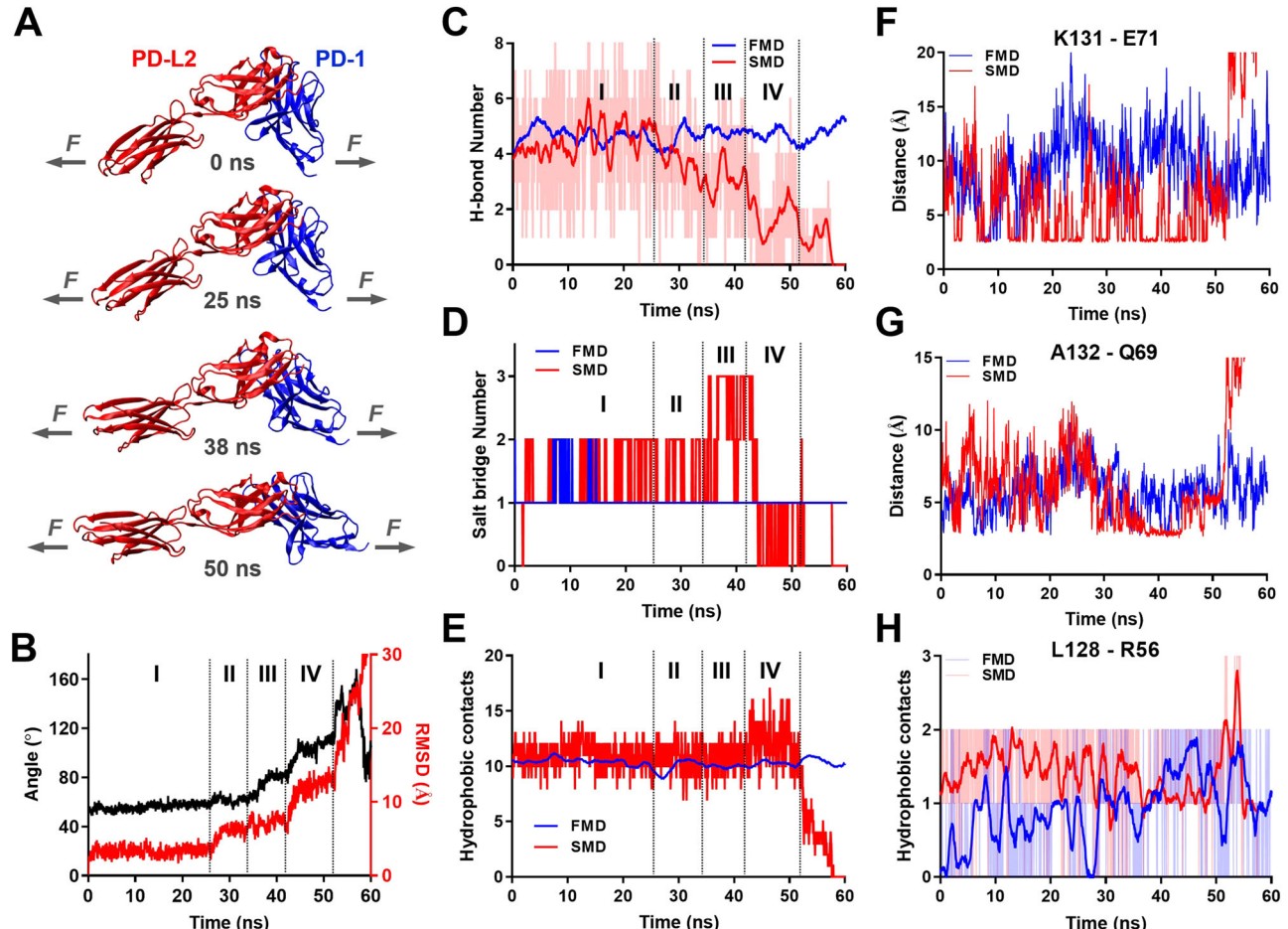

**Fig. 5 | Molecular dynamics (MD) reveals force induced PD-1–PD-L2 conformational change and formation of distinct atomic-level contacts.**
**A** Snapshots of PD-1–PD-L2 complex undergoing conformational changes in response to force at indicated simulation times. **B** Changes in relative angle (black curve, left y-axis) and root mean square displacement (RMSD, red curve, right y-axis) between PD-1 and PD-L2 in response to force. Comparison of total number of hydrogen bond (H-bond, **C**), salt bridge (**D**), and hydrophobic contacts (**E**) between PD-1 and PD-L2 during free MD (FMD, blue) and force steered MD (SMD, red). **F**–**H** Comparison of dynamics of putative interactions between indicated residues of PD-1 and PD-L2 during FMD (blue) and SMD (red). Atomic-level contacts were defined by an interatomic distance of <3.5 Å, which were more frequently observed in SMD than in FMD. Source data are provided in Source Data file.

## PD-1 mutants with impaired PD-1–PD-L2 mechanical stability show reduced inhibitory function

Combining the data of Figs. 1, 2 prompts us to hypothesize that the inhibitory function of PD-1 requires mechanical support of PD-Ligand to counterbalance the force the cell exerts on the PD-1 bonds. To test this hypothesis, we investigated whether disrupting PD-1–PD-L2 mechanical stability would affect PD-1 function. Using the NFκB::eGFP reporter Jurkat cells we tested the ability of PD-1 WT and different mutants to suppress TCR-CD3 triggered GFP induction by coculturing them with TSC cells with or without PD-L2 (Fig. 7A). Compared to control TSC not expressing PD-L2 (TSC-CTRL), TSC-PD-L2 cells triggered robust suppression of GFP induction in reporter cells expressing WT PD-1 relative to plain cells not expressing PD-1 (Fig. 7B–D). The inhibitory effect was significantly reduced by all three PD-1 mutants with the level of impairment ranking as K131 < L128A/K131A < A132K (Fig. 7B–D), which are the same in cell spreading and tension measurements (Fig. 6E–J). Together, these data suggest that mechanical forces on PD-1–PD-Ligand bonds critically regulate PD-1 ligand interaction and function.

## Discussion

Despite the success of immune checkpoint blockade of PD-1 or PD-L1 in cancer immunotherapy, the molecular mechanism of PD-1 triggering as to how ligand binding translates into biochemical signaling remains poorly understood. Dissecting the structural, biophysical, and biochemical bases of this mechanism is important as it would expand our fundamental understanding of signal initiation of a general class of transmembrane receptors – the enzyme-linked receptors – and guide the design of therapeutic agents targeting these receptors. While formation of PD-1 microclusters after ligand binding and their colocalization with TCR and CD28 were observed and found critical for PD-1's inhibitory function, other studies suggest that such colocalization is not required as HEK cells expressing PD-L1 is able to suppress the activating signal of bead-coated anti-CD3 and anti-CD28 on T cells in a co-culture system[11]. Consistent with the latter finding, our observation that bead-coated PD-Ligands were able to suppress in Jurkat cell NFκB activation induced by soluble anti-CD3 or primary T cell spreading and calcium flux induced by pMHC coated on spatially separated location also support the contention that localized PD-1 signaling is able to suppress TCR-CD3 signaling globally. A deep dive into the triggering process also begs the question as to whether the microcluster formation and co-localization drive PD-1 signal initiation or instead are merely their consequence. What seemingly have made this more perplexing are the findings that soluble isoforms of PD-L1 or PD-L2 in patients of different cancers fail to suppress T cell function in vitro and in vivo despite their ability to bind PD-1[9–11,30,31]. Comparing the same ectodomain of PD-L1 and PD-L2 in the form of soluble tetramer to that immobilized on bead surface, we further confirmed that

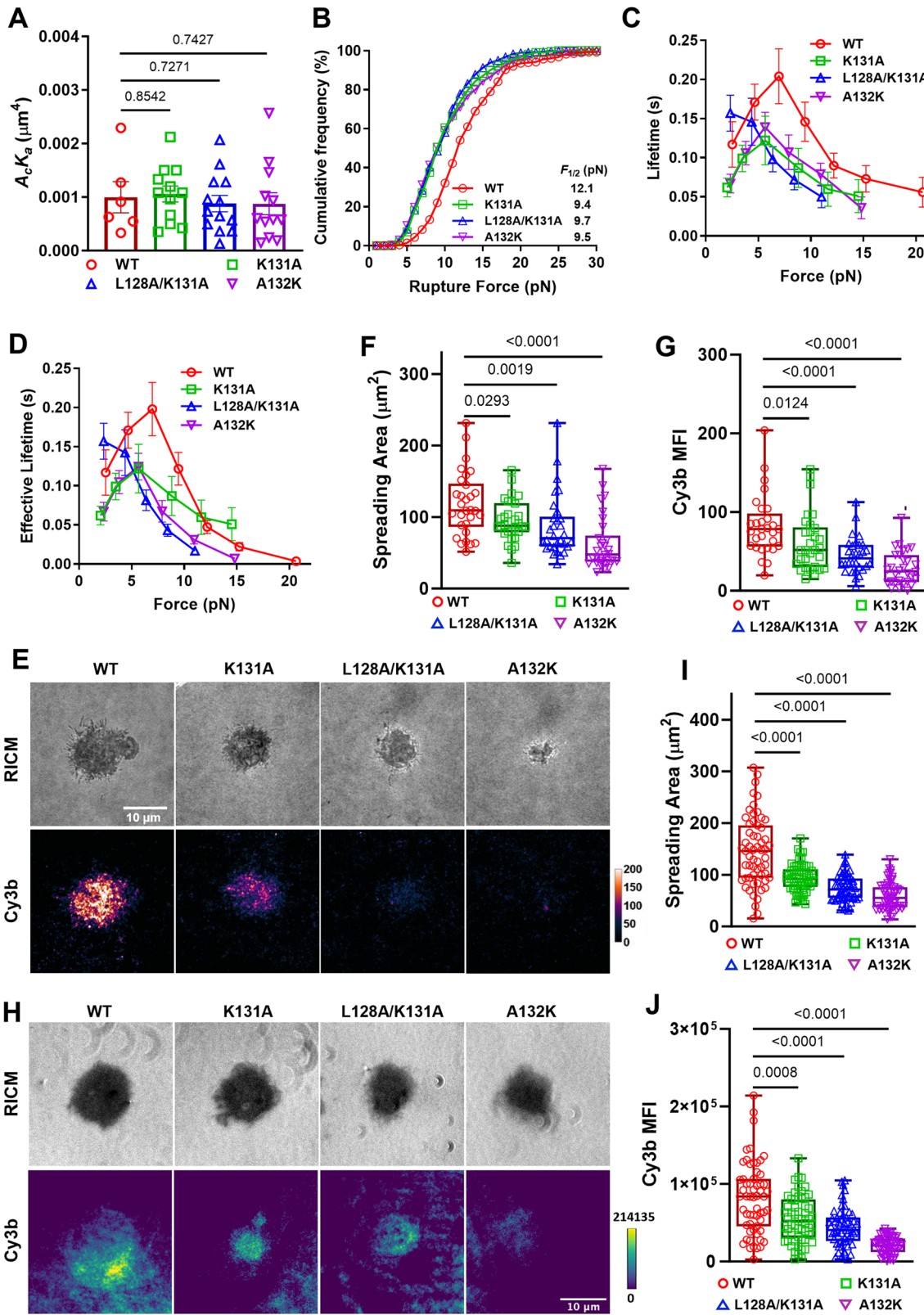

surface anchor of the PD-Ligand enhances PD-1 function, an important factor that has been largely ignored. Moreover, the inefficiency of multimeric ligand binding to trigger PD-1 signaling suggests that crosslinking of PD-1 by soluble PD-Ligand is not robust to initiate its signaling, and the formation of microclusters are likely a consequence of ligand binding (and signaling) instead of a mechanism of triggering.

What, then, does the bead-coated PD-Ligand provide but soluble PD-Ligand cannot? Built upon a recent study[13], we found that CHO cells, Jurkat cells, or activated primary T cells actively apply mechanical forces on PD-1–PD-L1 or PD-1–PD-L2 bonds. The endogenous forces exerted on PD-1 may result from PD-1's intracellular coupling to cytoskeleton, as ligand-bound PD-1 forms microclusters and moves centripetally on T cell membrane[5,14], and expressing PD-1 and PD-L2 in

**Fig. 6 | PD-1 mutants preventing force-induced atomic contacts impair PD-1–PD-L2 mechanical stability. A** Mean ± sem and individual measurements of 2D effective affinity of PD-L2 binding to CHO cells expressing WT or indicated mutants of PD-1. $n$ = 6, 12, 13, and 12 cell pairs for WT, K131A, L128A/K131A, and A132K, respectively. **B** Cumulative frequencies of rupture force events for PD-L2 bonds with PD-1 WT ($n$ = 278 events), K131A ($n$ = 210 events), L128A/K131A ($n$ = 345 events), and A132K ($n$ = 270 events) expressed on CHO cells. $p < 0.0001$ comparing $F_{1/2}$ of WT and each mutant using two-tailed Mann–Whitney U test. **C** Mean ± sem bond lifetime vs force plots for single PD-L2 bonds with PD-1 WT ($n$ = 785 events), K131A ($n$ = 625 events), L128A/K131A ($n$ = 759 events), and A132K ($n$ = 780 events) on CHO cells. $p < 0.0001$ comparing lifetime vs force distributions of WT and each mutant using two-tailed two-dimensional Kolmogorov-Smirnov test. **D** Effective bond lifetime calculated as multiplying bond lifetime in (**C**) by bond survival probability in (**B**). **E** Representative RICM and TIRF images of CHO cells expressing PD-1 WT or indicated mutants interacting with glass surface functionalized with PD-L2-coupled MTP of 4.7 pN threshold force (see Fig. S4A for schematic). Quantification of cell spreading area (**F**) and tension signal (**G**) for conditions in (**E**). $n$ = 29, 30, 30, and 30 pooled from 3 independent experiments. **H** Representative RICM and TIRF images of NFκB::eGFP reporter Jurkat cells expressing PD-1 WT or indicated mutants interacting with glass surface functionalized with PD-L2-coupled MTP of 4.7 pN threshold force (see Fig. 3A for schematic). Quantification of cell spreading area (**I**) and tension signal (**J**) for conditions in (**E**). $n$ = 59, 58, 59, and 59 cells from 1 in 2 independent experiments. Data were presented in box (median with 25%/75% boundaries) and whisker (min and max) plots. Numbers on graphs represent $p$ values calculated from two-tailed Mann–Whitney U test. Source data are provided in Source Data file.

CHO cells can drive the formation of synapse-like interface with concurrent accumulation of PD-1 and PD-L2[15]. This mechanical sampling process is quite intriguing because it suggests that the cell is actively probing the ON or OFF state of its membrane PD-1 molecules by exerting force on them, and those bound to an anchored ligand with mechanical support to counter-balance such forces respond to this probing and are recognized as in the ON state, whereas those without ligand binding or bound to soluble ligand are not or less able to be properly detected and thus more likely to remain in the OFF state. In other words, successful PD-1 triggering not only requires the formation of PD-1–PD-Ligand bonds, but is also enhanced when these bonds respond mechanically so as to be detected by the cells as a productive binding event. Structurally, this is also plausible, because there could hardly be any ON/OFF conformations defined from a molecule as simple as PD-1: a single IgV domain linked to intracellular domain by ~20 AA peptide of stalk and transmembrane region. In support of this hypothesis, we found that dampening the mechanical support of PD-Ligands by extending their length reduced the capacity of PD-1 to trigger function without affecting its PD-1 binding.

It follows from the above mechanical sampling hypothesis that the efficiency of PD-1 triggering is not merely determined by the PD-1 affinity for ligand as measured under force-free condition. In addition, the mechanical stability of the PD-1–PD-Ligand bonds – how the complex responds to force – would also be critical to the triggering process. This suggests bonds that readily rupture under force are less effective in initiating signals, whereas bonds that are more mechanically stable under force are more effective in triggering PD-1, implying the ability of PD-1 to sense the stiffness of the PD-Ligand expressing cell. Interestingly, we observed catch bonds for both PD-1–PD-L1 and PD-1–PD-L2 interactions, a phenomenon where increasing forces below an optimal level (7–8 pN for PD-1) promote bond stability instead of accelerating dissociation. This force-induced reinforcement of bond sustainability, or mechanical stability, is more profound for PD-1–PD-L2 than PD-1–PD-L1 interaction according to the rupture force distribution, catch bond profile, and MTP tension signal, suggesting a potential mechanism for ligand discrimination by PD-1. These results, although phenotypically similar to that observed in TCR antigen recognition, differ in its underlying structural mechanisms. Our SMD simulations of PD-1–PD-L2 dissociation suggest that the unique positioning of the two IgV domains from PD-1 and PD-Ligand with "side-to-side" binding using relatively flat β-sheets makes the complex very sensitive to mechanical loads. Forces applied to the two C-termini of the complex may readily generate a "peeling" effect. In response to force, the two IgV domains may rotate and translate relative to each other, transitioning into a stretched and aligned conformation. More importantly, such large-scale rearrangement of the complex is likely coupled with formation of distinct atomic-level interactions, which were not observed under force-free FMD conditions. These in silico studies and the experimentally measured catch bond profiles suggest that mechanical stability of PD-1–PD-Ligand is critical to the mechanical sampling process of PD-1, without which PD-1 triggering may be

much less efficient as seen in the case of soluble ligand completely loosing mechanical support.

Indeed, comparing SMD vs FMD results we identified residues of PD-1 that were not in contact with PD-L2 in the crystal structure and in FMD but may form distinct interactions in SMD. Mutations aiming to prevent the formation of these distinct interactions did not alter the force-free PD-1–PD-L2 2D affinity, consistent with the assertion that they do not contribute to static binding, but significantly reduced the bond stability under force, as shown by lower rupture force, shorter and altered profile of bond lifetime, as well as reduced endogenous forces. Consequently, these mutants demonstrate impaired ability to trigger PD-1 function with the same rank-order as that in the tension signal. These results indicate force and mechanical stability of PD-1–PD-Ligand bonds play a critical role in the triggering of PD-1 signaling and function.

Overall, our data suggest a potential PD-1 triggering mechanism consisting of three key components: (1) mechanical sampling: cells apply forces on PD-1 to probe ligand binding; (2) mechanical support: pulling force from the PD-Ligand anchoring surface counterbalances the pulling force from PD-1; (3) mechanical stability: forces on PD-1–PD-Ligand bonds modulate their dissociation kinetics and triggering efficiency. Formation and movement of PD-1 microclusters upon ligand binding may be subsequent steps following such recognition event and provide mechanical feedback as PD-1 is being actively transported. Mechanistically, it is of great interest as to how mechanical information is integrated into this recognition process and finally leads to phosphorylation of PD-1. On the translational side, our results suggest modulating the mechanical support and mechanical stability of the PD-1–PD-Ligand system as a potential strategy to regulate PD-1 agonism for disease treatment.

## Methods
### Ethical statement
All experiments in this study were conducted following the protocols approved by the Institutional Review Board (IRB) and Institutional Care and Use Committee (IACUC) of the Georgia Institute of Technology.

### Proteins and antibodies
Mouse PD-L1 and PD-L2 with C-terminal biotin produced in CHO cells[34] were generous gifts of Dr. Simon J. Davis (University of Oxford, United Kingdom). PE-anti-PD-1 (clone RMP1-30, 1:20) and its isotype control PE-Rat IgG2b,κ (clone RTK4530, 1:20) were purchased from Biolegend. PE-anti-PD-L1 (clone MIH1, 1:20) and its isotype control PE-Rat IgG2a,λ (clone 557076, 1:20) together with PE-anti-PD-L2 (clone TY25, 1:20) and its isotype control PE-Rat IgG2a,κ (clone R35-95, 1:20) were purchased from BD Biosciences. APC-anti-CD45.2 (clone 104, 1:100) was purchased from Biolegend. Purified anti-CD3 (clone OKT3) and anti-mouse IgG2a (clone RMG2a-62) were purchased from Biolegend. Purified antiPD-1 blocking antibody (clone 29 F.1A12), biotinylated antiPD-1 antibodies (clone 29 F.1A12 and clone RMP1-30), and biotinylated isotype controls (Rat IgG2a,κ clone RTK2758 and Rat IgG2b,κ

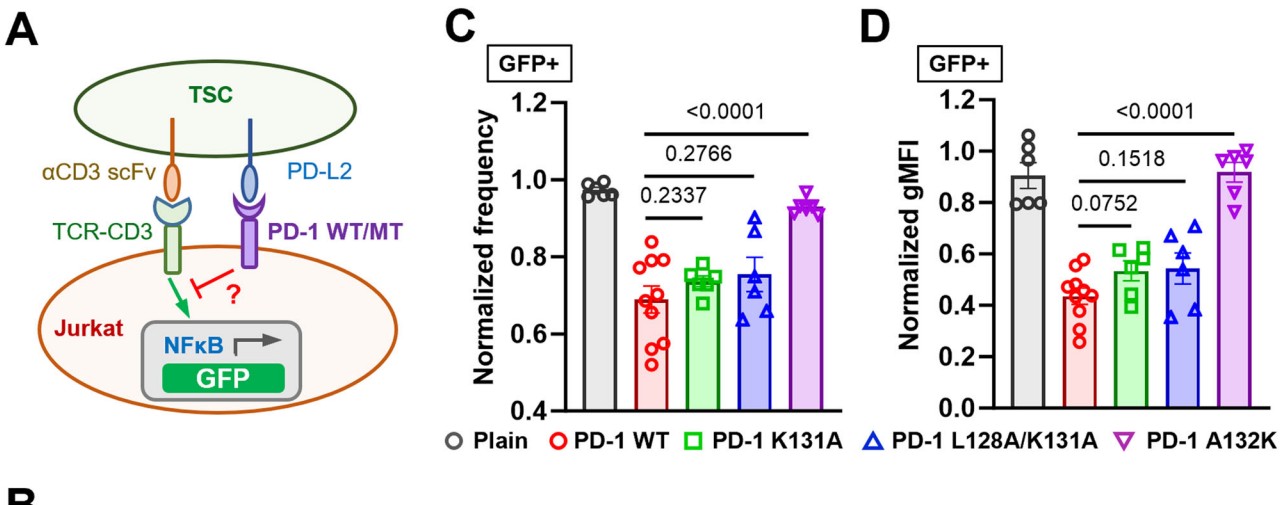

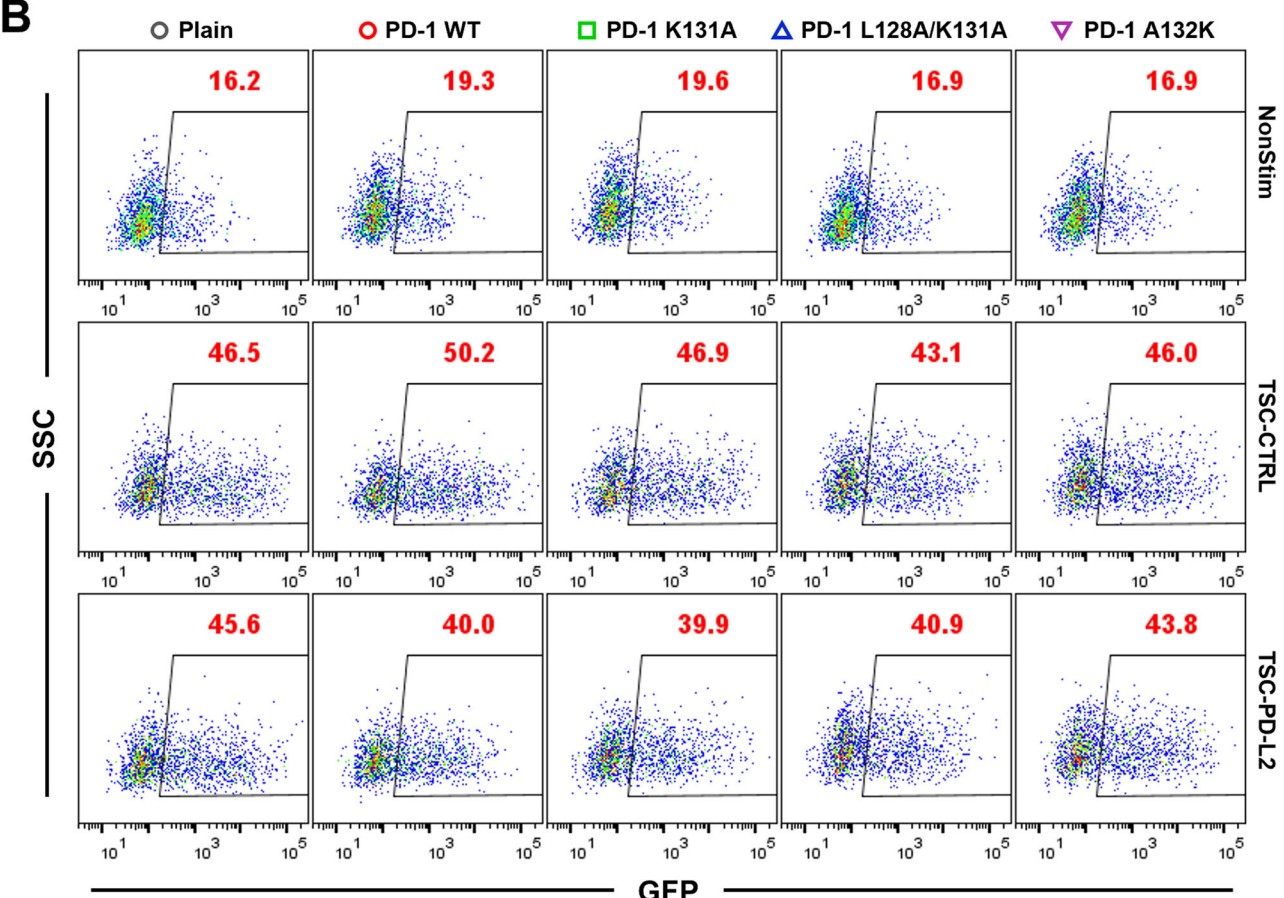

**Fig. 7 | PD-1 mutants with impaired PD-1–PD-L2 mechanical stability demonstrate reduced inhibitory function. A** Schematics of TSC-PD-L2 cells stimulation of NFκB::eGFP reporter Jurkat cells expressing no, WT, or indicated mutants of PD-1. **B** Representative SSC vs GFP plots of reporter Jurkat cells 24 h after stimulation with indicated conditions. **C**, **D** Quantification of GFP expression for conditions in (**B**). Normalized frequency (**C**) and normalized geometric mean fluorescence intensity (gMFI) (**D**) were calculated as (sample−averaged background)/(Plain−averaged background) and presented as mean ± SEM. *n* = 6 for plain, PD-1 K131A, L128K/K131A, and A132K pooled from 3 independent experiments or *n* = 10 for PD-1 reporter cells pooled from 5 independent experiments. Numbers on graphs represent *p* values calculated from two-tailed student *t* test. Source data are provided in Source Data file.

clone RTK4530) were purchased from Biolegend. Biotinylated antiPD-1 antibody (clone J43) and isotype control (clone eBio199Arm) were purchased from Invitrogen.

## Mice and cells
OT1 transgenic mice (C57BL/6-Tg(TcraTcrb)1100Mjb/J) were obtained from Charles River Laboratory (Lyon, France) and bred in house at the Georgia Institute of Technology. All mice were housed with the following conditions: temperature range between 20 and 26 °C, humidity of 40–70%, and a semi-natural light cycle of 12:12 light-to-dark ratio. Mice of both sexes (5 male and 3 females) aged 14-16 weeks old were used in this study. All protocols for sacrificing animals and isolation of splenocytes have been previously described[43]. To generate in vitro activated OT1 T cells, splenocytes from OT1 mice were pulsed with 10 nM of SIINFEKL peptide for 2 hrs, washed, and then cultured in RPMI 1640 supplemented with 10% FBS, 100 U/mL penicillin, 100 µg/

mL streptomycin, 2 mM L-glutamine, and 20 mM HEPES for 2–4 days. NFκB::eGFP reporter Jurkat cells and T-cell stimulator cells (TSC) expressing a membrane-anchored scFv of antiCD3 (clone OKT3) were generous gifts of Dr. Peter Steinberg (Medical University of Vienna, Austria). Jurkat cells, TSC, and CHO cells (ATCC cat. CCL-61) were cultured in RPMI 1640 supplemented with 10% FBS, 100 U/mL penicillin, 100 µg/mL streptomycin, 2 mM L-glutamine, and 20 mM HEPES. HEK 293 T cells (ATCC, cat. CRL-3216) were cultured in DMEM supplemented with 10% FBS, 6mM L-glutamine, 0.1mM MEM nonessential amino acids, and 1 mM sodium pyruvate. Human RBCs were isolated from healthy donors and used following previously described protocols[34] approved by the Institutional Review Board (IRB) of the Georgia Institute of Technology.

### Overexpression of PD-1 and PD-Ligands in CHO cells, Jurkat cells and TSC

All PD-1 mutants were generated using Q5 site-directed mutagenesis kit (NEB) following the manufacture's protocol. To generate CHO cells expressing WT or MT mouse PD-1, cells were transfected with pcDNA3.1 encoding full-length mouse PD-1 or its mutant using nucleofection (Lonza). Transfected cells were sorted for uniform PD-1 staining and culture in medium with 0.4 mg/ml G418. To generate CHO cells and reporter Jurkat cells stably expressing WT or MT chimeric PD-1 (mhPD-1) consisting of mouse PD-1 ectodomain (Met1 to Met169) and human PD-1 transmembrane and intracellular domain (Val171 to Leu288), full-length mhPD-1 were subcloned in lentiviral vector pLenti6.3. Lentivirus were produced by transfecting HEK 293 T cells with a mixture of mhPD-1-pLenti6.3, pMD2.G (Addgene #12259), and psPAX2 (Addgene #12260) using Lipofectamine 3000 (Thermo Fisher Scientific) following manufacture's protocol. CHO cells and reporter Jurkat cells were transduced overnight with 1:1 mixture of culture medium and lentiviral supernatant. Cells were then sorted for uniform and similar expression across WT and MT PD-1. To generate TSC expressing mouse PD-L1 or PD-L2, full-length PD-L1 or PD-L2 were subcloned into pMSCV-IRES-Thy1.1 (pMIT1.1) vector. Retrovirus were produced by transfecting HEK 293 T cells with a mixture of PD-L1-pMIT1.1 (or PD-L2-pMIT1.1) and pCL-Eco (Addgene #12371) using Lipofectamine 3000 (Thermo Fisher Scientific) following manufacture's protocol. TSC were transduced by spinoculation on retronectin-coated plate (Takara) with 1:1 mixture of culture medium and retroviral supernatant. Cells were subjected to repeated rounds of transduction and sorting to get desired PD-1 ligand expression.

### Flow cytometry

Cells were stained in 100 µl of FACS buffer (PBS supplemented with 5 mM EDTA and 2% FBS) containing fluorescently labeled antibodies (dilutions indicated above) for 30 min at 4 °C. After staining cells were washed twice with FACS buffer and analyzed using Fortessa flow cytometer (BD Biosciences) with FACSDiva v9. Flow cytometric data were analyzed using Flowjo v10 (TreeStar). Cells were gated based on FSC vs SSC for mono-population where expression of PD-1 and PD-Ligands were stained. In Jurkat and TSC coculture experiments, Jurkat cells were gated on CD45.2- population to exclude TSC.

### Stimulation of Jurkat cells

For experiments illustrated in Fig. S1D, 50,000 NFκB::eGFP reporter Jurkat cells were co-cultured with 50,000 TSC-CTRL, TSC-PD-L1, or TSC-PD-L2 for 24 h. After stimulation, cell mixture was stained with APC-antiCD45.2, washed, and analyzed by flow cytometry. For experiments illustrated in Figs. 1A, 1D and S2, 50,000 NFκB::eGFP reporter Jurkat cells were stimulated with 10 µg/ml anti-CD3 (clone OKT3) and 5 µg/ml anti-mouse IgG2a,κ secondary antibody for 24-30 hrs. For soluble PD-Ligand (or antibody) groups, a final concentration of 20 µg/ml pre-made PD-Ligand (or antiPD-1 antibody or isotype Ig)

tetramer (concentration excluding SA) or the corresponding amount of SA were added. For bead-coated PD-Ligand (or antibody) groups, 3 µl of SA beads (Dynabeads M-280 Streptavidin) were coated with 300 ng of PD-Ligand (or antibody or isotype Ig) for 1 hr at room temperature. After coating, beads were washed twice with PBS and mixed with cells at 10:1 bead-to-cell ratio. After stimulation, cells were washed and resuspend in FACS buffer for flow cytometric analysis. For experiments illustrated in Fig. 1G, H, cells were stimulated with the same conditions as in Fig. 1D with 10:1 bead-to-cell ratio except for using glass beads that were activated with MPTMS (Sigma) followed by conjugation with SA using SMCC or SM[PEG]$_{24}$ crosslinker (Sigma). Due to the inter-experiment variation of baseline GFP expression and its induction, the absolute values of % GFP+ or its gMFI are not directly comparable across experiments. Therefore, we used the normalized frequency or gMFI to quantify the fold change of GFP expression relative to internal control groups. Normalized frequency and normalized geometric mean fluorescence intensity (gMFI) of GFP+ population were calculated as (sample−averaged background)/(anti-CD3 control−averaged background).

### DNA hairpin sequences

**A21B-Cy3B:** Cy3B – CGC ATC TGT GCG GTA TTT CAC TTT - /3Bio/ **SH-BHQ2:** /5-ThioC6-5/ - TTT GCT GGG CTA CGT GGC GCT CTT - /3BHQ_2/ **12 pN HP:** GTG AAA TAC CGC ACA GAT GCG TTT GGG TTA ACA TCT AGA TTC TATTTT TAG AAT CTA GAT GTT AAC CCT TTA AGA GCG CCA CGT AGC CCA GC **4.7 pN HP:** GTG AAA TAC CGC ACA GAT GCG TTT GTA TAA ATG TTT TTT TCA TTT ATA CTTTAA GAG CGC CAC GTA GCC CAG C. **4.7 pN Locker strand:** AAA AAA CAT TTA TAC **12 pN Locker strand:** AAT CTA GAT GTT AAC CC

### AuNP DNA tension sensor preparation

AuNP-based tension probes were prepared following our previous work[13,36,44]. No. 2 glass coverslips (VWR: 48382085) were sonicated for ~5 min in nanopure water (18.2 MΩ) followed by ~5 min of sonication in pure ethanol. Coverslips were dried at 80 °C for 10 min and then etched in a fresh piranha solution containing 37.5% v/v hydrogen peroxide (30% solution) and 62.5% v/v concentrated sulfuric acid. Coverslips were then washed six successive times in nanopure water, followed by three successive washes in pure ethanol. Coverslips were then immersed in a 3%v/v APTES (Sigma-Aldrich: 440140) solution in ethanol for one hour at room temperature. Following this reaction, coverslips were washed three times with ethanol and baked in an 80 °C oven for 30 min. After cooling, the silanized coverslips were incubated with 1% w/v lipoic acid polyethylene glycol-succinimidyl ester (Biochempeg: HE039023-3.4 K, MW 3400) and 10% w/v monofunctional polyethylene glycol-succinimidyl ester (Biochempeg: MF001023-2K, MW 2000) in a freshly prepared 0.1 M NaHCO33 solution for at least 1 reacted hour at room temperature. Following PEG functionalization, coverslips were washed with nanopure water and blocked with a 1% w/v solution of Sulfo-NHS-Acetate (Thermo: 26777) in a freshly prepared 0.1 M NaHCO33 solution for 30 min to neutralize the positive charges of the amines and prevent nonspecific DNA binding to the surface. Following blocking, coverslips were washed with nanopure water and incubated with 400 µL of a 20 nM solution of 8.8 nm gold nanoparticles (AuNPs, nanoComposix) for 30 min. Finally, the DNA tension probe hairpins were assembled in 1 M NaCl by mixing Cy3B-labeled A21B (0.33 µM), BHQ2 strand (0.33 µM), and hairpin strand (0.3 µM) in a 1.1: 1.1: 1 ratio. The solution was heated to 95 °C and cooled to 25 °C in a thermocycler over a period of 30 min. After heating and cooling, 2.7 µM of passivating BHQ2 ssDNA was added to the DNA hairpin solution. Following DNA assembly, AuNPs were rinsed off the coverslips with nanopure water followed by rinsing with a 1 M NaCl solution. Coverslips were placed in a petri dish and 100 µL of the DNA hairpin was added to the surface. A second AuNP-functionalized surface was placed upside-down on top of the first coverslip to create a "sandwich."

Sandwiched slides were incubated with DNA overnight at 4 °C and further functionalized the following day.

## DNA hairpin ligand functionalization

DNA sandwiched coverslips were removed from 4 °C and separated into single coverslips. Coverslips were washed in 1X PBS and 200 μL of 40 μg/mL streptavidin (Rockland: S000-01) solution was added to each. Streptavidin was incubated on the coverslips for 1 h. Following incubation, coverslips were washed with 1X PBS and further functionalized with C-terminal-biotinylated mPDL1 or mPDL2 at a concentration of 15 μg/mL and placed in imaging chambers containing 1 mL of cellular imaging media (1x HBSS with 10 mM HEPES).

## Fluorescence Imaging of cell spreading and tension

For experiments in Fig. 6E–G and S4, imaging was done on a Nikon Eclipse Ti microscope, operated by Nikon Elements software, a 1.49 NA CFI Apo 100x objective, perfect focus system, and a TIRF laser launch with a 80 mW 561 nm laser. A reflection interference contrast microscopy (RICM) cube (Nikon: 97270) was used for imaging. An X-Cite 120 lamp (Excelitas) was used for widefield epifluorescence illumination. Images were acquired with an Andor iXon Ultra 897 electron-multiplying charge-coupled device. Fluorescent images acquired with TIRF excitation were taken with 100 ms exposure time, 10% laser power, and no gain. For experiments in Figs. 2A–C, 3, 6H–J, and S3, imaging was performed using Nikon W1 spinning disk confocal microscope equipped with a Plan-Apochromat 60x/1.40 oil objective, an RICM module for cell spreading, and a TIRF mode for fluorescent images. For experiments in Fig. 2, locker strand was added at 200 nM at t ~ 10 min to map the tension track for 10 min.

## Fluorescence Imaging of calcium flux

For experiments in Fig. 2D–F, in vitro activated OT1 T cells loaded with the calcium indicator X-Rhod-1, AM were pre-incubated with bead-coated or soluble PD-L1/L2 tetramer (or BSA) and washed. Cells with PD-L1/L2 beads were placed on surface functionalized with SIIN-FEKL:H2-K$^b$. Cells pre-incubated with BSA beads and tetrameric PD-L1/L2 were also placed on the same surface with soluble PD-L1/L2 continuously present. Calcium imaging was performed with 580 nm excitation and 602 nm emission on a cell-by-cell basis from the moment when the cell touched down on the surface by sedimentation and continued for 25 min at 37 °C.

For experiments in Fig. 2G, H, individual in vitro activated OT1 T cells loaded with the calcium indicator Fura-2 was aspirated by a micropipette (Fig. 2G bright field image, left) in a cell chamber mounted on the stage of an inverted microscope with temperature controlled at 37 °C. A small micropipette (Fig. 2G bright field image, lower right) was used to aspirate a bead coated with PD-L1/L2 or BSA, brought it to touch the cell from the side, and held it there. The chamber media contained either BSA if the beads were coated with PD-L1/L2 or tetrameric PD-L1/L2 if the beads were coated with BSA. A human red blood cell (RBC) coated with SIINFEKL:H2-K$^b$ aspirated by another pipette (Fig. 2G bright field image, right) was axially aligned with the left pipette and driven by the programed piezoelectric motor to contact the T cells in repeated cycles (each cell pair was tested for 200 repeating cycles, where they contacted for 0.2 sec per cycle). The intracellular calcium fluxes induced by the repeated intermittent TCR–pMHC interactions were measured by ratiometric imaging of 510 nm emission at 340 nm/380 nm excitations for more than 300 s.

## Micropipette adhesion frequency assay

As previously described[34,45–47], the 2D effective affinity between PD-L2 and WT or MT PD-1 were measured by analyzing the bond formation between PD-L2-coated human RBCs and target cells expressing WT or MT PD-1. In brief, RBCs were biotinylated, coated with SA, washed, and then coated with biotinylated PD-L2. During experiments, a PD-1 expressing cell was repeatedly brought into contact with a PD-L2-coated RBC, held for 5 seconds, and then separated. Due to the ultra-soft spring constant of the RBC membrane, PD-1–PD-L2 bonds formed during the contact (and last till separation) caused stretch of RBC membrane upon separation, and therefore defined as an "adhesion" event. The process was repeated for 30–50 cycles per RBC-Target cell pair and an averaged adhesion frequency ($P_a$) was calculated. $P_a$ is related to the 2D effective affinity of the molecular interaction in question according to the following equations:

$$P_a = 1 - \exp(-<n>) \tag{1}$$

and

$$<n> = m_r m_l A_c K_a [1 - \exp(-k_{off} t_c)] \tag{2}$$

Here $<n>$ is the average number of bonds per contact, $m_r$ and $m_l$ are the respective densities of PD-1 on target cell and PD-L2 on RBC, $A_c$ is contact area (in μm$^2$), $K_a$ is 2D affinity (in μm$^2$), and $k_{off}$ is off-rate (in s$^{-1}$). With long contact duration such as 5 s used in these experiments, $k_{off} t_c \gg 1$[47], $P_a$ and $<n>$ approach equilibrium, and the 2D effective affinity $A_c K_a$ was estimated by normalizing $<n>$ against $m_r$ and $m_l$ that were measured using PE-labeled monoclonal antibody together with QuantiBRITE PE standard beads (BD Biosciences):

$$A_c K_a = <n> / m_r m_l \tag{3}$$

## Biomembrane force probe

A previously described[20,34] BFP was used to measure the rupture force and bond lifetime at certain clamp force levels of PD-1–PD-Ligand bonds. A target cell expressing PD-1 is repetitively brought into contact with a PD-L1- or PD-L2-coated glass bead attached to a micropipette aspirated RBC. The displacement of the bead is tracked at 1000 fps with nanometer precision and is then translated into force trace with a BFP spring constant preset to 0.3 pN/nm. Compression of RBC during approaching and contact phase generate negative force values, whereas tension on the molecular bond formed between the ligand on the bead and the receptor on the target cell pulls the bead away from its resting position during target cell retraction. For rupture force measurement, the cell was retracted continuously after each contact. If a bond formed the force would increase linearly until bond rupture at which point the force level was recorded as the rupture force value. Hundreds of bond rupture events from repeated cycles were pooled to construct the rupture force histogram and the cumulative frequency of rupture events. For bond lifetime measurement, the target cell was initially retracted and then held at a distance corresponding to the preset clamp force level. The force sustained until the bond ruptured, with the total duration defining the bond lifetime under the corresponding clamped force level. Hundreds of bond lifetime events were pooled from repeated measurements in a range of clamp forces, which were used to construct the bond survival frequencies at different force binds and to calculate average bond lifetime vs average force by binning the events across multiple force levels. Data was collected and analyzed using Labview (v2016 and v2019).

## Molecular dynamics simulation

To bridge the gap between experimental observations and the underlying molecular interactions, all-atom MD simulations were performed. The crystal structure of mPD-1–mPD-L2 (PDB code: 3BP5) was used as the initial coordinates of the atoms. No membrane environment was included since the primary focus of the studies was on the binding and dynamics between the ectodomains of PD-1 and PD-L2. The protein structure was placed and solvated in a rectangular box of TIP3P water molecules with a 15 Å minimal distance between

boundaries and proteins imposed to avoid protein interacting with their mirror images. Explicit solvent can capture the formation and disruption of H-bonds, the screening of electrostatic interactions, and the involvement of solvent-mediated interactions. Water molecules were neutralized by adding $Ca^{2+}$, $Cl^-$, and $Na^+$ ions to create ~50 mM calcium concentration and ~100 mM ionic concentration that are physiologically relevant. The system had in total ~200,000 atoms (including 1 $Ca^{2+}$, 66 $Cl^-$, and 66 $Na^+$) in a water box of $160 \times 78 \times 60$ Å³. The x-axis was increased to accommodate the steered simulations and warranty that the protein complex was always inside the simulation box to avoid interaction with mirror images. Free molecular dynamics (FMD) and steered molecular dynamics (SMD) simulations[48] were performed using the NAMD program[49] and all topology and parameter files were generated using the CHARMM27 all-atom force field for proteins[50]. Fixed-angle force fields reasonably represent the non-covalent interactions in receptor–ligand binding while speeding up the simulations. The SHAKE[51] was used to constrain bond lengths involving bonds to hydrogen atoms. Periodical boundary condition was used along with particle mesh Ewald method for electrostatic interaction with a grid spacing of 1.0 Å and a 12-Å cutoff for van der Waals interaction. The time-step employed in all simulation was 2 fs with a 12 Å non-bonded cutoff and with long-range non-bonded interactions evaluated every two steps. The systems were initially energy-minimized with a conjugate gradient method for three stages of 50,000 steps each: all atoms of the proteins fixed, then only backbone atoms fixed, and finally all atoms free. After energy minimization, the system was heated up from 0 to 300 K in 300 ps and then equilibrated for 1 ns with pressure and temperature control. The temperature was held at 300 K using Langevin dynamics and the pressure was held at 1 atm by Langevin piston method. During the equilibration step, the root-mean-square deviations (RMSDs) of all Cα atoms reached a plateau and fluctuated around ~4 Å, indicating that equilibrium had been reached. After this step, FMD was performed on the equilibrated system for 100 ns to simulate an environment without mechanical force input. Then using the results of FMD simulations, different initial structures were chosen for running SMD to mimic the presence of mechanical forces on the PD-1–PD-L2 interactions. In all production SMD simulations, the controls on pressure and temperature were turned off to reduce disturbance on the systems. For SMD, the $C_\alpha$ atom of Met136 of PD-1 domain was pulled through a spring with a spring constant of 70 pN/Å at a constant speed of 1 Å/ns. The $C_\alpha$ atom of Leu229 of PD-L2 was constrained and held fixed to its equilibrated position. The results are independent of the initial conditions since the results were consistent across different simulations with different parameters and initial configurations sampling the relevant space. Simulation data were recorded at 50 ps steps unless stated otherwise. Distinct interaction formation was defined using the interatomic distance following criteria: a hydrogen bond is defined between an atom that has a hydrogen bond (the Donor) and another atom (the Acceptor) if their distance is less than the cut-off distance (3.2 Å) and the angle Donor-H-Acceptor is less than the cut-off angle (20°); a salt bridge is created if the distance between any of the oxygen atoms of acidic residues and the nitrogen atoms of basic residues is within the cut-off distance (3.5 Å) in at least one frame. The principal moments of inertia of the protein were used to measure the angle between interacting domains. VMD[52] (v1.9.3) was employed to analyze simulations, render molecular graphics, and generate trajectory videos. A total of ~1 μs of simulations were performed, including 1 FMD and 9 SMD simulations.

## Statistical analysis
Statistics comparing the mean values of two groups were calculated based on two-tailed student-*t* test or Mann–Whitney U test as indicated in figure legends. Statistics comparing two lifetime vs force distributions were calculated based on two-tailed two-dimensional

Kolmogorov-Smirnov test. Statistics were calculated using Microsoft Excel 2016, GraphPad Prism v10, and Matlab 2020a.

## Reporting summary
Further information on research design is available in the Nature Portfolio Reporting Summary linked to this article.

## Data availability
Data supporting the findings of this study are presented in the article and supplementary materials and are available from the corresponding authors upon request. Source data are provided with this paper.

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

## Acknowledgements

We thank Simon J. Davis for providing the mouse PD-L1 and PD-L2 with C-terminal biotin produced in CHO cells and Peter Steinberg for providing the NFκB::eGFP reporter Jurkat cells and T-cell stimulator cells (TSC). The MD simulations were supported by an NSF award (MCA08X014) in advanced computing infrastructure for U.S. and performed in the Extreme Science and Engineering Discovery Environment (XSEDE). This work was supported by NIH grants R01CA243486 (to C.Z.), U01CA250040 (to C.Z. and R.A.), U01CA250040S2 (to C.Z. and V.E.W.), RM1GM145394 (to K.S.) and NIH NCI F31 grant F31CA243502 (A.V.K.).

## Author contributions

K.L. and C.Z. conceived and coordinated the project. K.L., P.L., J.L., A.V.K., M.L., P.C., V.E.W., Z.Y., E.A., L.D., and Z.L. conducted the experiments and analyzed the data. K.L., K.S., R.A., and C.Z. wrote and revised the manuscript.

## Competing interests

The authors declare no competing interests.
