## [Peer Review File · Nature Communications]

Reviewers' Comments:

Reviewer #1:

Remarks to the Author:

This paper investigates the potential PD-1 triggering mechanism by employing active forces between cells, offering fresh insights into the intricate interactions within the synapse. However, the current findings fall short of providing full substantiation for the presented statements. The paper can be accepted if the authors are able to address the following questions.

(1) Despite observing the interaction between PD1 and PD-L1/PD-L2 under normal conditions, it's imperative to account for the modified circumstances of the tumor microenvironment (TME). How do the mechanical behaviors governing the PD1-PD-L1 interaction in the TME, including conditions like low pH, differ from those in normal conditions?

(2) Considering that TCR antigen recognition is intertwined with force application and dynamic response as well, it is essential to elucidate how the mechanical aspect links PD-1 signaling with TCR signaling. Can this analysis potentially provide insights into any clinical observations?

(3) Based on the current results, can the lifespan of PD1-PD-L1 bonds/interactions be quantitatively evaluated across different quantities of bonds?

(4) Additionally, it is advisable to provide further elucidation for Figure 2. Enhancing clarity by addressing the data of 4.7 pN+PD-1 and resequencing the sub-figures would be beneficial.

Minor:

(1) Please check and confirm the P value in Fig S2G.

(2) Please make the description of "PD-1" consistent throughout the manuscript, such as the wrong description in Fig S2C "PD1".

(3) Fig 5F, please correct the description of "PD-1 WT".

(4) Fig 4 C-H, please make the style of the line symbols of FMD/SMD consistent among these figures.

(5) Please clarify whether these residues (Leu128, Lys131 and Ala132) are the key residues those contact with PD-L1 and PD-L2 according to their crystal structures.

Reviewer #2:

Remarks to the Author:

Li et al. argue that the application of mechanical force to the activation marker PD-1 plays a pivotal role in PD-1's co-inhibitory function in T cells. The authors base their assertion on a series of experimental approaches, many of which have previously been employed by the senior author in the context of studying TCR-peptide/MHC interactions. These approaches include:

1. Stimulation of Jurkat-GFP reporter cells through engineered T cell stimulator cells or soluble means (e.g., antibodies, PD-L1/-L2 monomers, multimers, coated beads).
2. Force measurements conducted on PD1-expressing CHO cells interacting with surfaces presenting PD-L1 or -L2 tethered through force-responsive DNA-probes.
3. Frequency adhesion assays under defined force conditions.
4. Molecular dynamics simulations.
5. Mutagenesis of PD-1L.

The evidence presented in the manuscript is suggestive and far from conclusive, primarily due to the utilization of inherently noisy and artificial cellular systems and the absence of adequate control experiments.

Key Concerns:

1. Choice of Jurkat cells and functional readout: Drawing functional conclusions from experiments using Jurkat cells is challenging, if not infeasible, due to significant alterations in signaling pathways stemming from the absence of PTEN. This concern is particularly pertinent in the context of this study, given that co-stimulation, affected by PD-1 triggering, is a central focus. Unlike T cells, Jurkat cells exhibit atypical characteristics, such as high levels of plasma membrane PIP3, which is typically associated with CD28-driven co-stimulation, a target of PD-1. Jurkat cells proliferate without the need for antigenic or co-stimulatory inputs (in contrast to primed T cells), a phenomenon heavily dependent on the absence of PTEN. Notably, Jurkat cells display spurious signaling in the absence of antigen and appear inherently insensitive to presented antigen, raising uncertainties about the suitability of this model. Figures 1 and 6 underscore these concerns, as a substantial proportion of assayed Jurkat cells upregulate NFAT-driven GFP even without antigen stimulation, and less than 50% display GFP expression despite the presence of antigen. Consequently, the observed level of PD-1-driven inhibition of T cell activation appears relatively minor. Given these limitations, it is doubtful whether this system can yield definitive insights into the mechanisms underpinning PD-1 co-inhibition based on the subtle changes in GFP expression following varied treatments.

2. Comparisons between different modes of stimulation: Comparing the outcomes of

cross-linking TCRs with aggregated anti-CD3 antibodies with T-cell stimulations involving surface-resident antigen is problematic, especially when assessing the impact of co-inhibitory modalities. Since the mechanisms governing TCR triggering differ, it is surprising that the authors draw conclusions from this experimental setup (comparing TCS vs. soluble antibody). This approach leaves a fundamental question unanswered regarding the extent to which observed differences result from the surface-bound or soluble modes of T cell stimulation or co-inhibition. Without directly experimentally affecting co-inhibitory modalities within the same T cell: APC system, it is challenging to extract meaningful insights.

3. Choice of CHO Cells for force sensing: The choice of CHO cells expressing hamster PD-1 in experiments involving force sensing (Figure 2, 3, and 5) raises questions about the appropriateness of this model. It is unclear why murine T cells, a more logical choice, were not employed. Furthermore, it remains unclear how these highly artificial CHO cell-based experimental conditions inform our understanding of the frequency of catch bonding in PD-1:PD-L1/2 interactions within the immune synapse. Given the premise of this manuscript, it should be expected that force shielding within the immune synapse (e.g. via CD2:CD58, LFA-1:ICAM-1 or other adhesion systems) should significantly impact the scale of PD-1:PD-L1/2-imposed forces and, consequently, the co-inhibitory effect of PD-1. Demonstrating such behavior would provide a highly convincing argument.

4. Overstating the relevance of Molecular Dynamics Simulations (MDS): Molecular dynamics simulations should be regarded as a hypothesis-generating tool, not source of definitive facts. The authors' presentation of simulation results as conclusive findings is not aligned with scientific principles.

5. Lack of controls: The study neglects the crucial aspect of assessing potential 3D binding effects resulting from PD-L1 mutations, which could have been addressed using methods such as surface plasmon resonance or similar techniques.

Additional Points:

1. Variations in ligand densities: It should be considered that the densities of stimulatory and co-inhibitory ligands differ across various stimulation modes. Quantitative measurements could help elucidate the qualitative distinctions observed.

2. Data presentation: All 2D cytometric plots lack tick marks and annotations.

3. Clarity of figures: Figure 1 lacks clarity regarding structure and the description of experimental regimens. What setting refers to what data point in the diagrams? It took

me considerable time to grasp the content of the figures.

4. Flow cytometric readout: The flow cytometric readout concerning T cell stimulation (i.e. GFP expression) is oversimplified and presented as “normalized frequency”. The reader’s perception of the ms. will likely benefit from less data processing, especially in view of the noisy nature of the GFP-based measurements.

Reviewer #3:

Remarks to the Author:

In this manuscript, Li et al. investigated the mechano-sensing properties of PD-1 and its ligands. They provided thorough evidence to support that mechanical force is critical for PD-1 triggering. This novel discovery is critical for understanding the mechanism of PD-1 signaling. Using model cell lines, they showed PD-1 signal was not triggered by soluble ligands without any mechanical support. The conclusion that T cells apply forces toward the ligand was further supported by the force measurement using DNA-based molecular tension probes (MTPs), biomembrane force probe (BFP), and Molecular dynamics simulation. Using PD-1 mutants of the same binding affinity but with lower rupture forces, they demonstrated that reduced mechanical interaction impaired the suppression function of PD-1. The authors have use multiple technologies to thoroughly characterize the mechanical interaction between PD-1 and its ligand. The conclusions are in general supported by the experimental results. The discovery of the mechanical nature of PD-1-ligand interaction could potentially provide insight into the biomarker discovery in the clinic for immunotherapies.

The following issues have to be addressed before publication:

1. Fig 1: to further prove that T cell forces are necessary, the authors should apply cytoskeleton inhibitors to block T cell forces and compare the suppression effect on GFP.
2. Fig I-K: PEG is not a good choice for elongation as it forms random coil in solution. An alternative (probably better) way is to elongate the receptors on T cell surface, such as the methods reported in these manuscripts: *J Immunol* June 1, 2010, 184 (11) 5959-5963, and *Front. Immunol.*, 10 July 2017.
3. In Fig 2 and 3: it is not clear why the authors switched to CHO cells rather than Jurkat cells? why not use primary T cells, such OT-1 T cells, with which the result would much better reflect the real situation.
4. Fig 6A: the “Plain” with TSC-ctrl should be shown.
5. In general, validation using primary T cells is lacking. For example, to confirm the impaired suppression function of soluble ligands (or elongated ligands), primary T cells

can be used.

6. It remains unclear how T cells exert forces through PD-1 and its ligands. Any evidence to show the connection of PD-1 with cytoskeleton? If so, how?

7. A minor issue: page 5, "Figs. 1A-B" should be "Fig. 1A-B". same mistakes in the whole paragraph when referring to Fig. 1.

Response to Reviewers' comments:

Reviewer #1:

(1) Despite observing the interaction between PD1 and PD-L1/PD-L2 under normal conditions, it's imperative to account for the modified circumstances of the tumor microenvironment (TME). How do the mechanical behaviors governing the PD1-PD-L1 interaction in the TME, including conditions like low pH, differ from those in normal conditions?

We thank the Reviewer for asking this question, which is important given the success of PD-1 blockade for cancer therapy. To address this concern and concerns of Reviewer 2, we developed several new experiments to investigate the cross-talks between signaling by TCR and PD-1 (please see Fig. R4 in the response to Reviewer 2 for schematics of one such experiment). In these experiments, we compared primary T cells spreading on and calcium signaling induced by pMHC-coated surfaces while in contact with PD-L1/L2 either coated on beads or as tetramer in solution. The results of one such experiment are shown in Fig. R1 where the spreading areas of in vitro activated OT1 T cells on SIINFEKL:H2-K^b surface in fresh media (pH = 7.76) vs. media used to culture B16F10 melanoma cells for 4 days (pH = 7.02) were compared. As suspected by the Reviewer, we indeed observed that cell spreading was significantly reduced in a couple of conditions by the tumor-conditioned media compared to the normal media (Fig. R1). However, the pattern that the bead-coated, but not soluble tetrameric, PD-L1/2 reduce T cell spreading on pMHC were observed in both media (Fig. R1), indicating that using tumor-conditioned media did not negate the observation made in normal media that mechanical support is important for the inhibitory function of PD-1.

Despite the positive outcome, these results did not fully address the question of how the mechanical behaviors governing the PD-1–PD-L1 interaction in the TME differ from those in normal conditions, which we respectfully submit that are questions beyond the scope of the present paper for many reasons. For one, although tumor conditioned media contain factors that may be present in the tumor microenvironment such as low pH (actually, both 7.76 and 7.02 are within normal pH range of cell media), it still does not fully mimic TME, which is also highly variable depending on the tumor type and stage. In addition, PD-1 plays a much broader and more fundamental role in mediating peripheral tolerance in healthy and a variety of disease settings such as autoimmunity and (chronic) viral infection, where TME is irrelevant. Even in the case of solid tumor, a large part of the response to PD-1 blockade originates from stem-like CD8⁺ T cells in lymph nodes instead of in the TME. Therefore, the significance

Fig. R1. Melanoma cell line-conditioned media did not negate the observation made in normal media that mechanical support of PD-ligand is important for PD-1's inhibitory function. Representative images by reflection interference contrast microscopy (A) and quantification (B) of activated OT1 CD8⁺ T cells spreading on SIINFEKL:H2-K^b coated surface in contact with beads coated with PD-L1, PD-L2, or BSA in the presence of tetrameric PD-L1, PD-L2, or BSA in solution. The experiment was done in either normal media or B16F10 melanoma cell-conditioned media. Black numbers on graphs represent p values calculated from two-tailed Mann-Whitney U test of indicated two groups or PD-Ligand groups (green or blue) with BSA control (black). Red numbers on graphs represent p values calculated from two-tailed Mann-Whitney U test of normal media vs B16F10 media for each group.

of our findings includes their relevance to the general PD-1 mechanobiology and is not limited to TME of solid tumors. For these reasons, we only add the data in Fig. R1 as Supplementary Figure S3A&B in the revised manuscript, and interpret it conservatively.

Related to the review's question re pH, previous studies suggest that the interaction of PD-1 with its ligands is dominated by hydrophobic contacts and not sensitive to pH changes¹.

(2) Considering that TCR antigen recognition is intertwined with force application and dynamic response as well, it is essential to elucidate how the mechanical aspect links PD-1 signaling with TCR signaling. Can this analysis potentially provide insights into any clinical observations?

We thank the Reviewer for this suggestion, as we also recognize the importance of elucidating the linkage between PD-1 signaling and TCR signaling. In fact, the present manuscript is our second paper on this topic; the first is also published in *NComms* (Li et al. 2021, PD-1 suppresses TCR-CD8 cooperativity during T-cell antigen recognition)². To further elucidate the mechanical aspects of the links between PD-1 signaling and TCR signaling, we have performed an additional experiment in which the experimental configuration was inverted, i.e., OT1 T cells in contact with SIINFEKL:H2-K^b (or BSA as control) coated on beads or in solution (as tetramer) were placed on surfaces functionalized with PD-L1/L2 (or BSA as control) in the absence (for the pMHC beads) or presence (for the BSA beads) of OVA pMHC tetramer in solution. We measured the spreading area of the T cells on PD-L1/L2 while the cells' TCR was also stimulated by pMHC on beads or in solution. We found that the spreading areas of OT1 T cells activated in vitro to express PD-1 were significantly higher on PD-L1/L2 than on BSA, which were statistically indistinguishable between conditions in which T cells were concurrently stimulated by pMHC on beads, in solution (as tetramer) or unstimulated (Fig. R2). Note that such results are in sharp contrast to those of an experiment with the original configuration (Fig. R1). Despite this interesting data, we wish to be more conservative in its interpretation. Cell spreading on PD-L1/2 may not represent PD-1 signaling in its full extent. We respectfully submit that to conclude that the cross-talk between the TCR and PD-1 signaling is unidirectional, such that PD-1 mechanosignaling inhibits TCR mechanosignaling, but not the other way around, would require more extensive studies that are beyond the scope of the present paper. Therefore, in the revised manuscript we only included two experiments similar to Fig. R1 in normal media (one shown in Fig. R1 as side-by-side control to B16F10 conditioned media) in the main body (new Fig. 2A-C) but Fig. R2 in Supplementary Fig. 3C.

Fig. R2. Engagement of TCR with bead-bound or tetrameric pMHC in solution had no effect on T cell spreading on PD-L1/L2-coated surface. Activated OT1 T cells spreading on PD-L1, PD-L2, or BSA surface in contact with SIINFEKL:H2-K^b or BSA coated beads in the absence or presence of tetrameric SIINFEKL:H2-K^b in solution. Numbers on graphs represent p values calculated from two-tailed Mann-Whitney U test.

(3) Based on the current results, can the lifespan of PD1-PD-L1 bonds/interactions be quantitatively evaluated across different quantities of bonds?

The data in new Fig. 4G (old Fig. 3G) show the respective average lifespans of single PD-1 bonds with PD-L1 and PD-L2 in forces ranging from 2.5-20 pN. The data in the new Fig. 6D and 6E (old Fig. 5D and 5E) show the respective average lifespans of single bonds of wild-type and three mutant PD-1 molecules with PD-L2 in the same force range. To use these results to quantitatively evaluate different quantities

of bonds requires information regarding how the applied force on the cell is shared by the different bonds mediating the adhesion. Using the simplest assumption that the force is equally shared by all the bonds, then from the force F we can calculate the average lifespan for the first bond that fails from the curves in new Figs. 4G, 6D and 6E (depending on the interaction of interest) at the force of F/n (n = the total number of bonds). Next, we can use the same curves but take the lifetime evaluated at the force of $F/(n - 1)$ to evaluate the average lifespan for the second bond that fails. This process continues until we reach the point that there is no more bond left. The sum of all these average lifespans can be taken as a rough estimate for the average lifespan of an n -bond adhesion. We emphasize that the calculation depends on four assumptions: 1) the applied force remained constant during the dissociation process, 2) this force is equally shared by all remaining bonds at every step when one bond fails, 3) bonds fail sequentially as opposed concurrently, and 4) after one bond fails, the remaining bonds reset their clock for their lifespans. While these simplifying assumptions seem reasonable for getting a first approximation to the desired answer, at present there is no way to prove or falsify their validity based on the available technology. For this reason, we feel that it would be too speculative to include the above discussion in the manuscript, which would distract the readers from focusing on the central theme of the paper.

(4) Additionally, it is advisable to provide further elucidation for Figure 2. Enhancing clarity by addressing the data of 4.7 pN+PD-1 and resequencing the sub-figures would be beneficial.

We thank the Reviewer for this suggestion. In response and to address concerns of Reviewer 2, we have moved the old Fig. 2 to Fig. S4 and replace it with new data obtained using primary OT1 T cells in vitro activated to express PD-1 (Fig. R3) in Fig. 3 of the revised manuscript.

Fig. R3. Activated OT1 T cells spread on and exert forces to PD-L1/L2 conjugated with molecular tension probes (MTP). **A.** Schematic showing the working principle of MTA. **B.** Representative images by reflection interference contrast microscopy (RICM, upper row) of activated OT1 T cells spreading on PD-L1 or PD-L2 conjugated 4.7 or 12 pN MTP in the absence or presence of anti-PD-1 antibody and the corresponding Cy3b fluorescence imaged by total internal reflection fluorescence microscopy (TIRF, lower row). **C.** **D.** Quantification of the spreading area (C) and the Cy3b fluorescence (D) illustrated in B. Numbers on graphs represent p values calculated from two-tailed Mann-Whitney U test.

Minor:

(1) Please check and confirm the P value in Fig S2G.

Done.

(2) Please make the description of "PD-1" consistent throughout the manuscript, such as the wrong description in Fig S2C "PD1".

Done.

(3) Fig 5F, please correct the description of "PD-1 WT".

Done.

(4) Fig 4 C-H, please make the style of the line symbols of FMD/SMD consistent among these figures.

Done.

(5) Please clarify whether these residues (Leu128, Lys131 and Ala132) are the key residues those contact with PD-L1 and PD-L2 according to their crystal structures.

The answer to Reviewer's question is YES, as stated in the original manuscript: "In particular, we noticed that some of the force enhanced atomic contacts were not located in the binding pocket or disrupt force-free PD-1–PD-L2 binding when mutated³, such as Leu128, Lys131 and Ala132 located in the FG loop of PD-1 (Figs. 4F-H)," To further clarify this point, we added the phrase "of their crystal structure" before the end of this sentence.

Reviewer #2 (Remarks to the Author):

1. Choice of Jurkat cells and functional readout: Drawing functional conclusions from experiments using Jurkat cells is challenging, if not infeasible, due to significant alterations in signaling pathways stemming from the absence of PTEN. This concern is particularly pertinent in the context of this study, given that co-stimulation, affected by PD-1 triggering, is a central focus. Unlike T cells, Jurkat cells exhibit atypical characteristics such as high levels of plasma membrane PIP3, which is typically associated with CD28-driven co-stimulation, a target of PD-1. Jurkat cells proliferate without the need for antigenic or co-stimulatory inputs (in contrast to primed T cells), a phenomenon heavily dependent on the absence of PTEN. Notably, Jurkat cells display spurious signaling in the absence of antigen and appear inherently insensitive to presented antigen, raising uncertainties about the suitability of this model. Figures 1 and 6 underscore these concerns, as a substantial proportion of assayed Jurkat cells upregulate NFAT-driven GFP even without antigen stimulation, and less than 50% display GFP expression despite the presence of antigen. Consequently, the observed level of PD-1-driven inhibition of T cell activation appears relatively minor. Given these limitations, it is doubtful whether this system can yield definitive insights into the mechanisms underpinning PD-1 co-inhibition based on the subtle changes in GFP expression following varied treatments.

We thank the reviewer for the comment. We agree that it is important to choose the proper biological system to evaluate PD-1's function and that Jurkat has its own genetic variations of certain signaling molecules. However, Jurkat cells remain a well-established model system in the field and our stimulation does not involve the CD28 axis. The reporter Jurkat cells were developed and validated by Prof. Peter Steinberger of the Medical University of Vienna to provide a robust and easily adapted system for assaying the effects of co-stimulatory and/or co-inhibitory molecules on TCR signaling⁴. These cells have been used by several groups successfully (personal communications from Dr. Michelle Krogsgaard and Dr. Jun Wang, both of NYU). Therefore, we used these Jurkat reporter cells to test whether the

upregulation of GFP in Jurkat cells, driven by NF κ B signaling (not NFAT) upon anti-CD3 antibody stimulation, was reduced by treatment with bead-coated or soluble PD-Ligand.

Nevertheless, we understand the Reviewer's concerns and agree that our main conclusion, that PD-1

Fig. R4. Schematics and representative images of the first two experiments comparing the inhibitory function of bead-coated vs soluble PD-L1/L2 on T cell calcium signaling. A. In vitro activated OT1 CD8⁺ T cells loaded with the calcium indicator X-Rhod-1 were pre-incubated with bead-coated or soluble PD-L1/L2 tetramer (or BSA) and washed. Cells with PD-L1/L2 beads were placed on surface functionalized with SIINFEKL:H2-K^b. Alternatively, cells pre-incubated with BSA beads and tetrameric PD-L1/L2 were also placed on the same surface with soluble PD-L1/L2 continuously present. Calcium imaging was performed with 578 nm excitation and 600 nm emission on a cell-by-cell basis from the moment when the cell touchdown on the surface by sedimentation and continued for 25 min at 37 °C.

signaling is enhanced by mechanical support because it allows the cell to exert endogenous forces on PD-1–PD-L1/L2 bonds, should be confirmed using primary T cells. Instead of repeating all experiments in the paper using primary T cells, which would require the very challenging task of mutating PD-1 in primary T cells, we designed three sets of experiments. In the first two sets of experiments, we examined 1) spreading of, 2) calcium signaling in primary OT1 CD8⁺ T cells which were put in contact with PD-L1/L2 (or BSA for control) coated beads (15- or 30-min prior incubation and presence during the experiment) or in solution (as tetramer) and then placed on SIINFEKL:H2-K^b (or BSA for control) coated surface in the absence (for the pMHC-beads) and presence (for the BSA-beads) of solution PD-L1/L2 tetramer (Fig. R4). Previously, we used the first two experiments in our 2021 *NComm* paper (Li et al. PD-1 suppresses TCR-CD8 cooperativity during T-cell antigen recognition)² where P14 T cells were placed on surfaces co-immobilized with gp33:H2-D^b and PD-L1/L2 to show that PD-1 signaling inhibited TCR signaling-mediated spreading and calcium fluxes. Here we separated the presentation of pMHC and PD-L1/L2 on two different surfaces.

In the third experiment, we employed the fluorescence micropipette adhesion frequency assay previously used in our 2014 *Jl* paper to show that repeatedly forming and breaking bonds of SIINFEKL:H2-K^b coated red blood cells with OT1 TCR would induce T cell calcium signaling⁵. Here we added a third micropipette to aspirate a bead coated with PD-L1/L2 (or BSA as control) to contact the T cells in the absence (for PD-L1/L2 bead) or presence (for BSA bead) of PD-L1/L2 tetramer in solution (Fig. R5).

Our results of the first two experiments show that the spreading areas on pMHC surface (see Fig. R1 in response to Reviewer 1's comment), or calcium fluxes (Fig. R6A), of primary T cells in contact with beads coated with PD-L1/L2 (with

Fig. R5. Photomicrograph of the third experiment comparing the inhibitory function of bead-supported vs soluble PD-L1/L2 on T cell calcium signaling. A. An activated OT1 CD8⁺ T cell loaded with the calcium indicator Fura-2 was aspirated by the left micropipette in a cell chamber mounted on the stage of an inverted microscope with temperature control at 37 °C. A small micropipette from the lower right was used to aspirate a bead coated with PD-L1/L2 or BSA, bring it to touch the cell from the side, and hold it there. The chamber media contained either BSA if the beads were coated with PD-L1/L2 or tetrameric PD-L1/L2 if the beads were coated with BSA. A human red blood cell (RBC) coated with SIINFEKL:H2-K^b aspirated by another right pipette was axially aligned with the left pipette and driven by the programed piezoelectric motor to contact the T cells in repeated cycles (each cell pair was tested for 200 repeating cycles, where they contacted for 0.2 sec per cycle). The intracellular calcium fluxes induced by the repeated intermittent TCR–pMHC interactions were measured by ratiometric imaging in the fluorescence channel of the microscope for > 300 s. B. Representative pseud-color 340/380 ratio image of a T cell fluxing calcium. See Supplementary Video.

BSA in solution) were significantly lower than those T cells in contact with beads coated with BSA (with or without tetrameric PD-L1/L2 in solution) in all except one case ($p = 0.066$ for comparison of calcium fluxes inhibited by bead coated vs soluble PD-L2, Fig. R6A). By comparison, T cells in contact with BSA-bearing beads are mostly statistically indistinguishable in the presence and absence of tetrameric PD-L1/L2 in solution (Figs. R1 and R6A). For the third experiment, bead coated, but not solution, PD-L1 significantly suppressed calcium, which is similar to the first two experiments. However, both PD-L2 coated on the beads and in solution significantly suppressed calcium (Fig. R6B). These results are consistent with those obtained using the Jurkat cells, supporting the validity of our conclusion and the usefulness of the reporter cell system in the present work despite the potential issues of the Jurkat cells as suggested by the Reviewer. However, we also found that soluble PD-L2 could also suppress calcium signaling. As a result, we softened our statement from “mechanical support is required...” to “mechanical support enhances...” We have added these new results to the revised manuscript (new Fig. 2).

Fig. R6. Mechanical support of PD-L1/L2 is important for them to inhibit calcium signaling in T cells induced by pMHC. **A.** Data from experiments shown in Fig. R4 for comparison of calcium fluxes in activated OT1 CD8⁺ T cells induced by SIINFEKL:H2-K^b coated surface in contact with bead coated vs solution PD-L1/L2. **B.** Data from experiments shown in Fig. R5 for imaging of calcium in single activated OT1 CD8⁺ T cells induced by repeated contact cycles of a SIINFEKL:H2-K^b coated RBC while the T cell was also in contact with a bead coated PD-L1/L2 in comparison with the case where the T cell was also in contact with a bead coated BSA and the chamber solution contained tetrameric PD-L1/L2. Numbers on graphs represent p values calculated from two-tailed Mann-Whitney U test.

2. Comparisons between different modes of stimulation: Comparing the outcomes of cross-linking TCRs with aggregated anti-CD3 antibodies with T-cell stimulations involving surface-resident antigen is problematic, especially when assessing the impact of co-inhibitory modalities. Since the mechanisms governing TCR triggering differ, it is surprising that the authors draw conclusions from this experimental setup (comparing TCS vs. soluble antibody). This approach leaves a fundamental question unanswered regarding the extent to which observed differences result from the surface-bound or soluble modes of T cell stimulation or co-inhibition. Without directly experimentally affecting co-inhibitory modalities within the same T cell: APC system, it is challenging to extract meaningful insights.

We appreciate the concern raised by the Reviewer. In fact, our experiments presented in the original manuscript considered this point by comparing the inhibitory function of PD-1 between two settings 1) **soluble anti-CD3 stimulation** and soluble PD-L1/L2 tetramer (illustrated in Fig. 1A) and 2) **soluble anti-CD3 stimulation** and bead-coated PD-Ligand (illustrated in Fig. 1D). In both cases, T cells were stimulated in the same way. The only difference is how PD-ligand is presented. The first dataset using TSC serves the general purpose to show that the reporter Jurkat can be used to study PD-1 function in our system. Hence, we respectfully submit that we did not draw the main conclusion based on comparison between TSC simulations and soluble anti-CD3 stimulations.

Nevertheless, the Reviewer’s concern has been addressed by the additional experiments performed in response to Critique #1 above (Figs. R4-6), which were performed using specific pMHC to stimulate primary T cells. The new results support the same conclusion, which is that PD-1 signaling is enhanced by mechanical support that allows the cell to exert endogenous on PD-1–PD-L1/L2 bonds.

3. Choice of CHO Cells for force sensing: The choice of CHO cells expressing hamster PD-1 in experiments involving force sensing (Figure 2, 3, and 5) raises questions about the appropriateness of this model. It is unclear why murine T cells, a more logical choice, were not employed. Furthermore, it remains unclear how these highly artificial CHO cell-based experimental conditions inform our understanding of the frequency of catch bonding in PD-1:PD-L1/2 interactions within the immune synapse. Given the premise of this manuscript, it should be expected that force shielding within the immune synapse (e.g. via CD2:CD58, LFA-1:ICAM-1 or other adhesion systems) should significantly impact the scale of PD-1:PD-L1/2-imposed forces and, consequently, the co-inhibitory effect of PD-1. Demonstrating such behavior would provide a highly convincing argument.

We recognize that real immunological synapses involve many different molecular interactions. However, we are not aware of any publications supporting the “force shielding” effect of adhesion molecules. To the contrary, data published by the Salaita lab show that LFA-1–ICAM-1 interaction enhances the forces on TCR–pMHC bonds instead of “shielding” them⁶. In this first study of PD-1 mechanoimmunology, we took a reductionist approach and used model systems to investigate the mechanical regulation of PD-1–PD-L1/L2 and TCR–pMHC interactions separately from other complicating factors, such as CD2–CD58 and LFA-1–ICAM-1 interactions, which will be included in future studies. Nevertheless, the Reviewer’s point is well taken. In response, we performed new experiments using primary OT1 T cells (see Fig. R3 from response to Critique #4 of Reviewer 1) and Jurkat cells transduced with wild-type and three mutant PD-1 molecules (Fig. R7). Our results are consistent with data obtained using CHO cells. In the revised manuscript, we have presented these new data in the main body of the paper and moved the CHO cell data to Fig. S4.

Fig. R7. Jurkat T cells expressing wild-type (WT) or indicated mutant PD-1 were compared for their abilities to spread on and exert forces to PD-L1/L2 conjugated with molecular tension probes (MTP). A. Representative images by reflection interference contrast microscopy (RICM, upper row) of Jurkat T cells spreading on PD-L1 or PD-L2 conjugated 4.7 pN MTP and the corresponding Cy3b fluorescence imaged by total internal reflection fluorescence microscopy (TIRF, lower row). B, C. Quantification of the spreading area (B) and the Cy3b fluorescence (C) illustrated in A. Numbers on graphs represent p values calculated from two-tailed Mann-Whitney U test.

4. Overstating the relevance of Molecular Dynamics Simulations (MDS): Molecular dynamics simulations should be regarded as a hypothesis-generating tool, not source of definitive facts. The authors' presentation of simulation results as conclusive findings is not aligned with scientific principles.

We have softened the language in the revised manuscript.

5. Lack of controls: The study neglects the crucial aspect of assessing potential 3D binding effects resulting from PD-L1 mutations, which could have been addressed using methods such as surface plasmon resonance or similar techniques.

We showed 2D affinity measurements of the PD-1 mutants (old Fig. 5B and new Fig. 6A). Our previous publication showed that the 2D and 3D measurements of the PD-1–PD-L1/L2 interactions are well correlated⁷.

Additional Points:

1. Variations in ligand densities: It should be considered that the densities of stimulatory and co-inhibitory ligands differ across various stimulation modes. Quantitative measurements could help elucidate the qualitative distinctions observed.

Receptor and ligand site densities varied from different experiments because primary T cells, Jurkat cells, and CHO cells were used in different experiments. Within each group of experiments, the site densities were well matched and controlled, but not always measured.

2. Data presentation: All 2D cytometric plots lack tick marks and annotations.

We have added tick marks and annotations in all 2D cytometric plots in the revised manuscript.

3. Clarity of figures: Figure 1 lacks clarity regarding structure and the description of experimental regimens. What setting refers to what data point in the diagrams? It took me considerable time to grasp the content of the figures.

The Reviewer's point is well taken. We have improved the figures in question in the revised manuscript.

4. Flow cytometric readout: The flow cytometric readout concerning T cell stimulation (i.e. GFP expression) is oversimplified and presented as "normalized frequency". The reader's perception of the ms. will likely benefit from less data processing, especially in view of the noisy nature of the GFP-based measurements.

We acknowledge the Reviewer's suggestion. We have shown representative FACS plots of GFP expression in the original and revised manuscript. However, due to the inter-experiment variation of baseline GFP expression and its induction, the absolute values of % GFP+ or its gMFI are not directly comparable. What is important is the magnitude of GFP expression related to the internal control groups. Therefore, we used the normalized frequency or gMFI to quantify the fold change of GFP expression relative to control.

Reviewer #3 (Remarks to the Author):

In this manuscript, Li et al. investigated the mechano-sensing properties of PD-1 and its ligands. They provided thorough evidence to support that mechanical force is critical for PD-1 triggering. This novel discovery is critical for understanding the mechanism of PD-1 signaling. Using model cell lines, they

showed PD-1 signal was not triggered by soluble ligands without any mechanical support. The conclusion that T cells apply forces toward the ligand was further supported by the force measurement using DNA-based molecular tension probes (MTPs), biomembrane force probe (BFP), and Molecular dynamics simulation. Using PD-1 mutants of the same binding affinity but with lower rupture forces, they demonstrated that reduced mechanical interaction impaired the suppression function of PD-1. The authors have used multiple technologies to thoroughly characterize the mechanical interaction between PD-1 and its ligand. The conclusions are in general supported by the experimental results. The discovery of the mechanical nature of PD-1-ligand interaction could potentially provide insight into the biomarker discovery in the clinic for immunotherapies.

The following issues have to be addressed before publication:

1. Fig 1: to further prove that T cell forces are necessary, the authors should apply cytoskeleton inhibitors to block T cell forces and compare the suppression effect on GFP.

The Reviewer's point is well taken. However, we respectfully submit that treating the reporter cells with cytoskeleton inhibitors to see if GFP expression can be affected may not allow one to decisively rule in or rule out whether T cell forces on PD-1 are necessary for PD-1's inhibitory function because the GFP reduction (if observed) cannot be interpreted simply and solely by the inhibition of force on PD-1. TCR signaling also depends on the cytoskeleton so these inhibitors may affect activating signals via TCR and/or its interplays with PD-1 signaling, not necessarily solely affecting PD-1 signaling itself (TCR signaling can modulate PD-1 phosphorylation and thereby how efficiently it can signal). Also, given the low force amplitude on that PD-1 can sustain as shown in our MTP and BFP measurements, passive forces, instead of active forces from cytoskeleton, may account for a large part of mechanoregulation of PD-1. We believe that the experiments described in the original manuscript using GFP reporter cells and the new experiments in this revision using primary T cells (depicted in Figs. R4 and R5 and data shown in Figs. R1 and R6) have provided convincing evidence for our main conclusion. These data show that, for PD-1 to inhibit TCR's activation signal, it is important for PD-L1/L2 to anchor to a solid surface, which provides mechanical support to balance the endogenous forces exerted by T cells (which we have been shown in Figs. R3 and R7 for their existence).

Nevertheless, in an attempt to address the question raised by the Reviewer, we performed a different experiment to measure the effects of latrunculin A (inhibitor of actin polymerization), Y-27632 (ROCK inhibitor), blebbistatin (myosin II inhibitor), and NSC87877 (SHP1/2 inhibitor) on T cell endogenous forces (Fig. R8). Compared to the DMSO control, latrunculin A, Y-27632, blebbistatin, and NSC87877 (except for spreading) significantly reduced T cell spreading and forces on TCR engaged with pMHC. These results are consistent with reports of us⁸ and others⁶, and indicate that the forces on TCR are induced by pMHC engagement triggered signaling and powered by actin cytoskeleton and myosin motors. In sharp contrast, neither T cell spreading nor forces on PD-1 engaged with PD-L1/L2 were affected by any of the same inhibitors, despite the fact that the same batch of T cells and MTP surfaces were used to perform the experiment side-by-side with those for measuring TCR forces (Fig. R8). We can think of two possible explanations for these surprising results. First, even though we were clearly able to measure significant levels of specific T cell endogenous forces on PD-1 (compared to BSA control), their much lower levels than the TCR forces might limit our ability to discern the effects of the inhibitory agents (if any) due to the small dynamic range. However, this hypothesis is not supported by the fact that we were clearly able to distinguish the force differences on PD-1 when the MTP force threshold was changed from 4.7 pN to 12 pN (Fig. R3) and when the cell surface PD-1 was changed from WT to mutants (Fig. 7). Alternatively, the T cell forces on PD-1 depends on mechanisms other than SHP-

mediated signaling, actin cytoskeleton, and myosin motor. We respectfully submit that to identify such unknown mechanisms is a major research task that requires significant amount of time and efforts, hence should be left for future studies. Since we do not have a clear interpretation of the data in Fig. R8, we will not include them in the revised manuscript. They are shown here as privileged communication for reviewer's inspection only.

Fig. R8. T cells spreading and endogenous forces on TCR, but not on PD-1, were suppressed by pharmacological agents that inhibit actin polymerization, myosin II activity, and ROCK and SHP signaling. OT1 CD8⁺ T cells activated in vitro to express PD-1 were treated with the indicated inhibitors (or DMES control) and placed on surfaces functionalized with SIINFEKL:H2-Kb, PD-L1, PD-L2, or BSA conjugated with 4.7 pN MTP. Spreading area (A) and Cy3b fluorescence (B) were measured using RICM and TIRF, respectively. ns = not significant, *p<0.05, **p<0.01, ***p<0.0001 by Mann-Whiney U test.

2. Fig I-K: PEG is not a good choice for elongation as it forms random coil in solution. An alternative (probably better) way is to elongate the receptors on T cell surface, such as the methods reported in these manuscripts: J Immunol June 1, 2010, 184 (11) 5959-5963, and Front. Immunol., 10 July 2017.

We agree with the Reviewer that making fusion protein to elongate PD-1 would be an alternative and perhaps better approach, but that method would also significantly increase the complexity and the amount of work involved. On the balance of effort and yield, we believe our current method already made the point, because we already observed the predicted effect to support our conclusion, which is the bottom line. In an ongoing collaboration, we are using beads of variable viscoelastic properties to further explore the effect of changing mechanical support on PD-1 signaling. But these results are beyond the scope of the present work and will be presented in future papers.

3. In Fig 2 and 3: it is not clear why the authors switched to CHO cells rather than Jurkat cells? why not use primary T cells, such OT-1 T cells, with which the result would much better reflect the real situation.

The Reviewer's point is well taken. The reason is that we made the CHO cells earlier in the study to test whether the specific PD-1 mutations would alter catch bonds as suggested by the MD simulations. And we used them in the MTP experiment during these early days. The Jurkat cells were obtained in a later time for signaling studies after Prof. Peter Steinberger published their development and validation of

this reporter system⁴. In response, we repeated the experiment using primary T cells (OT1) and Jurkat cells. The new results are consistent with data obtained using CHO cells. In the revised manuscript, we have presented these new data in the main body of the paper and moved the CHO cell data to Supplementary Figures.

4. Fig 6A: the “Plain” with TSC-ctrl should be shown.

Fig. 6A (new Fig. S7B) shows PD-1 expression on Jurkat. It is not a co-culture experiment with TSC.

5. In general, validation using primary T cells is lacking. For example, to confirm the impaired suppression function of soluble ligands (or elongated ligands), primary T cells can be used.

The Reviewer’s point is well taken. In the revised manuscript we have included four additional experiments that used primary T cells: spreading on, calcium signaling (measured by two methods), and endogenous forces induced by pMHC coated surface. As can be seen in Figs. R1-R8, the new results confirm the conclusions obtained using CHO cells and Jurkat cells. The new data have been added to the revised manuscript.

6. It remains unclear how T cell exert forces through PD-1 and its ligands. Any evidence to show the connection of PD-1 with cytoskeleton? If so, how?

We agree with the Reviewer that the question of how T cells exert forces through PD-1 engaged with immobilized PD-Ligands is an important one, and at present we are not aware of any data showing PD-1’s connection with the cytoskeleton. We believe this will be an interesting topic for future studies as it is beyond the scope of the present work.

7. A minor issue: page 5, “Figs. 1A-B” should be “Fig. 1A-B”. same mistakes in the whole paragraph when referring to Fig. 1.

We thank the Reviewer for pointing out these mistakes. We have corrected them in the revised manuscript.

REFERENCES

1. Cheng, X. *et al.* Structure and interactions of the human programmed cell death 1 receptor. *The Journal of biological chemistry* **288**, 11771-11785 (2013).
2. Li, K. *et al.* PD-1 suppresses TCR-CD8 cooperativity during T-cell antigen recognition. *Nature communications* **12**, 2746 (2021).
3. Lazar-Molnar, E. *et al.* Crystal structure of the complex between programmed death-1 (PD-1) and its ligand PD-L2. *Proc Natl Acad Sci U S A* **105**, 10483-10488 (2008).
4. Jutz, S. *et al.* A cellular platform for the evaluation of immune checkpoint molecules. *Oncotarget* **8**, 64892-64906 (2017).
5. Pryshchep, S., Zarnitsyna, V.I., Hong, J., Evavold, B.D. & Zhu, C. Accumulation of serial forces on TCR and CD8 frequently applied by agonist antigenic peptides embedded in MHC molecules triggers calcium in T cells. *Journal of immunology* **193**, 68-76 (2014).

6. Liu, Y. *et al.* DNA-based nanoparticle tension sensors reveal that T-cell receptors transmit defined pN forces to their antigens for enhanced fidelity. *Proc Natl Acad Sci U S A* **113**, 5610-5615 (2016).
7. Li, K., Cheng, X., Tilevik, A., Davis, S.J. & Zhu, C. In situ and in silico kinetic analyses of programmed cell death-1 (PD-1) receptor, programmed cell death ligands, and B7-1 protein interaction network. *The Journal of biological chemistry* **292**, 6799-6809 (2017).
8. Hong, J. *et al.* A TCR mechanotransduction signaling loop induces negative selection in the thymus. *Nature immunology* **19**, 1379-1390 (2018).

Reviewers' Comments:

Reviewer #1:

Remarks to the Author:

The authors have addressed all the comments. I have no further concerns.

Reviewer #2:

Remarks to the Author:

The revised version of the manuscript by Li and colleagues has satisfactorily addressed all the points I raised in my previous assessment. The manuscript greatly benefits from showcasing the results of the additional experiments conducted in response to the reviewers' concerns. I fully endorse the publication of the current version.

One last point: the manuscript would benefit from including the reasons underlying the use of the term "normalized frequency" in the Methods (or Results) section, as explained in the authors' rebuttal letter under point 4 (Reviewer #2).

Reviewer #3:

Remarks to the Author:

All the previously mentioned issues have been well addressed. The conclusions are solid. The discovery of the mechanical nature of PD-1-ligand interaction is interesting and important for the future development of immunotherapies based on immune checkpoint inhibitors.